# VivosX, a disulfide crosslinking method to capture site-specific, protein-protein interactions in yeast and human cells

Chitra Mohan[1], Lisa M Kim[2], Nicole Hollar[2], Tailai Li[1], Eric Paulissen[1], Cheuk T Leung[2]*, Ed Luk[1]*

[1]Department of Biochemistry and Cell Biology, Stony Brook University, New York, United States; [2]Department of Pharmacology, University of Minnesota Medical School, New York, United States

**Abstract** VivosX is an *in vivo* disulfide crosslinking approach that utilizes a pair of strategically positioned cysteines on two proteins to probe physical interactions within cells. Histone H2A.Z, which often replaces one or both copies of H2A in nucleosomes downstream of promoters, was used to validate VivosX. Disulfide crosslinks between cysteine-modified H2A.Z and/or H2A histones within nucleosomes were induced using a membrane-permeable oxidant. VivosX detected different combinations of H2A.Z and H2A within nucleosomes in yeast cells. This assay correctly reported the change in global H2A.Z occupancy previously observed when the deposition and eviction pathways of H2A.Z were perturbed. Homotypic H2A.Z/H2A.Z (ZZ) nucleosomes accumulated when assembly of the transcription preinitiation complex was blocked, revealing that the transcription machinery preferentially disassembles ZZ nucleosomes. VivosX works in human cells and distinguishes ZZ nucleosomes with one or two ubiquitin moieties, demonstrating that it can be used to detect protein-protein interactions inside cells from different species.
DOI: https://doi.org/10.7554/eLife.36654.001

*For correspondence:
ctleung@umn.edu (CTL);
ed.luk@stonybrook.edu (EL)

**Competing interests:** The authors declare that no competing interests exist.

## Introduction

To uncover the mechanism of a molecular pathway, one approach is to understand how individual components interact within the physiological context of a cell. Existing cell-based, protein-protein interaction assays, such as yeast two-hybrid or protein-fragment complementation, do not provide information about the site of interaction or the proportion of free versus bound species (*Fields and Sternglanz, 1994*; *Michnick et al., 2007*). Here, we introduce a methodology called VivosX (*in vivo* disulfide crosslinking) which is a simple, quantifiable assay for reporting site-specific interactions that occur inside the nucleus or the cytosol. VivosX uses structural information to guide the placement of a pair of cysteine probes on the opposite sides of a contact site such that they fall within disulfide crosslinking distance when the two proteins interact. The disulfide adducts do not accumulate naturally due to the reducing environment of the nucleus and cytosol (*Dardalhon et al., 2012*; *Østergaard et al., 2004*), but will do so under oxidizing conditions. Interactions captured by disulfide crosslinking, along with any non-interacting, uncrosslinked species, are determined by non-reducing gel electrophoresis and immunoblotting. VivosX can be applied to detect the oligomerization status of nuclear and cytosolic factors based on the proportion of crosslinked and uncrosslinked species, providing a simple strategy to study, for example, how transcription factors or signaling molecules dimerize in response to cellular cues. As a proof of concept, VivosX was used to detect interactions between specific histone proteins within the nucleosomes of yeast and human cells.

The canonical view of the nucleosome structure, in which a ~150 basepair DNA wraps around an octameric histone core with two copies of each H2A, H2B, H3, and H4 (*Luger et al., 1997*),

represents only one of the many facets of the basic packaging unit of chromatin in a living cell (*Luger et al., 2012*). Nucleosomes in the vicinity of promoters, for example, are subjected to a variety of chromatin remodeling activities that alter not only nucleosome conformation, but also histone core composition (*Clapier et al., 2017*). When a chromatin remodeler repositions a nucleosome to facilitate transcription factor binding, it transiently stretches out a segment of the nucleosomal DNA at one region, makes a bulge at another, and distorts the histone core (*Deindl et al., 2013*; *Sinha et al., 2017*). To promote transcription of a gene, the nucleosome immediately downstream of an RNA polymerase II promoter is often installed with one or two copies of the variant histone H2A.Z in place of H2A (*Luk et al., 2010*; *Tramantano et al., 2016*; *Weber et al., 2014*). To understand transcriptional control in eukaryotes, and perhaps even in archaea (*Mattiroli et al., 2017*), it is important to decipher how histone movements are choreographed at different stages of transcription. Chromatin immunoprecipitation (ChIP)-based techniques have been instrumental in uncovering the dynamic interactions of a specific type or post-translationally modified histone with DNA (*Gilmour and Lis, 1984*; *O'Neill and Turner, 1995*). It remains technically challenging to distinguish nucleosomes containing different combinations of histones (e.g. homotypic H2A.Z/H2A.Z versus heterotypic H2A/H2A.Z nucleosomes). This obstacle motivated us to develop the VivosX technique to differentiate nucleosomes with distinct histone combinatorial states by capturing site-specific histone-histone interactions *in situ*. Importantly, this technique is not limited to studying histones, but can be used to examine many protein complexes within cells for which structural information is available.

Disulfide crosslinking of strategically positioned cysteines on reconstituted nucleosomes or endogenously purified histone substrates *in vitro* has generated important structural and molecular insights. For example, crosslinking of cysteine probes substituted at the basic N-terminal tail of H4 and the acidic patch of H2A demonstrated the direct interaction between the histone tail and the histone core of neighboring nucleosomes during chromatin fiber compaction (*Dorigo et al., 2004*). Disulfide crosslinking between two H3 molecules was used to demonstrate the tetrameric nature of the H3-H4 complex when bound by the histone chaperone Nap1 (*Bowman et al., 2011*). A disulfide crosslinking approach that restricts conformational flexibility of the histone fold domain of H3 and H4 revealed that histone fold distortion is a prerequisite of remodeler-catalyzed histone octamer sliding (*Sinha et al., 2017*). More recently, disulfide crosslinking has been applied to stabilize the conformation of nucleosomes to facilitate structural analysis (*Frouws et al., 2018*). The use of disulfide crosslinking to probe protein-protein interactions has also been fruitful for the studies of the conformational dynamics of transmembrane proteins, including chemoreceptors and rhodopsin (*Falke and Koshland, 1987*; *Farrens et al., 1996*).

While disulfide crosslinking has been used successfully *in vitro*, this approach has not been well developed as a general strategy for capturing protein-protein interactions in cells. One technical challenge is that the reducing environment inside the cytoplasm and nucleus inhibits disulfide accumulation even when cysteine pairs are within crosslinking distance (*Dardalhon et al., 2012*; *Østergaard et al., 2004*). Although cysteine pairs can be activated to form disulfide linkages in the presence of a thiol-reactive oxidant, such reactions are dependent on the accessibility of the thiol groups (*Johnson et al., 1987*). Thus, the choice of cysteine probe substitutions must be guided not only by structural information but also conformational dynamics.

Our group studies the molecular mechanisms that regulate the turnover of histone H2A.Z at promoters, a chromatin remodeling activity that is linked to a variety of chromosomal functions (such as transcriptional activation, chromosome segregation, DNA repair, and transcriptional anti-silencing), likely resulting from a general function in facilitating chromatin accessibility (*Dhillon et al., 2006*; *Krogan et al., 2004*; *Raisner et al., 2005*; *Rangasamy et al., 2004*; *Zhang et al., 2005*). At promoters of yeast and human cells, H2A.Z is inserted into +1 nucleosomes, which are located immediately downstream of the transcription start site (*Albert et al., 2007*; *Barski et al., 2007*) (*Figure 1— figure supplement 1*). In human cells, H2A.Z is also associated with heterochromatin (*Rangasamy et al., 2003*). Unlike H2A.Z found in euchromatin, heterochromatic H2A.Z is generally monoubiquitylated at the C-terminus (*Sarcinella et al., 2007*). What functional role the ubiquitin moiety has on H2A.Z remains unclear. SWR, a multi-subunit chromatin remodeler, catalyzes H2A.Z deposition via a histone exchange mechanism driven by ATP hydrolysis (*Mizuguchi et al., 2004*). SWR removes an H2A-H2B dimer and concomitantly inserts H2A.Z-H2B onto each face of the nucleosome, thereby converting the canonical H2A/H2A (AA) nucleosome to the heterotypic H2A/H2A.Z

(AZ) nucleosomal intermediate before forming the homotypic H2A.Z/H2A.Z (ZZ) nucleosomal product (*Luk et al., 2010*) (*Figure 1—figure supplement 1*). H2A.Z-containing nucleosomes are preferentially disassembled via a process that is dependent on the preinitiation complex (PIC), although it is unclear whether the ZZ or the AZ species are equally targeted for disassembly (*Tramantano et al., 2016*).

Within an AA, AZ or ZZ nucleosome, the two L1 loop regions of the histone fold domains of the opposing H2A and/or H2A.Z molecules come into contact with each other at an interface that passes through the nucleosomal dyad, but not when the histones are outside of the nucleosomal structure (*Luger et al., 1997*; *Suto et al., 2000*) (*Figure 1A*). Thus, the L1-L1' (prime represents the symmetrical nucleosomal counterpart) interface is ideal for the placement of cysteine probes for *in vivo* disulfide crosslinking. The assumption is that only when histones are inserted into nucleosomes will the cysteine pairs be close enough to form disulfide bonds, allowing crosslinking efficiency to act as a metric for the global occupancy of H2A and H2A.Z (these histones naturally lack cysteine).

In this study, we show that H2A-to-H2A, H2A.Z-to-H2A, and H2A.Z-to-H2A.Z crosslinking can be used to infer the levels of AA, AZ and ZZ nucleosomal species in yeast cells, demonstrating that VivosX can be used as a general strategy for probing interactions of other protein pairs in cells when structural information exists to allow strategic placement of the cysteine probes. The VivosX approach can be extended to probe the global configurations of H2A.Z of human cells. Interestingly, ZZ nucleosomes found in these cells are frequently monoubiquitylated on one H2A.Z molecule or symmetrically on both. The result highlights the usefulness of VivosX in determining the stoichiometry of bulky post-translational modifications on individual subunits of an oligomer.

## Results

### VivosX accurately captures intra-nucleosomal histone-histone interactions in yeast cells

To identify a cysteine substitution in the L1 region of H2A.Z that can crosslink to the same site on the opposite H2A.Z molecule within a ZZ nucleosome (*Figure 1A*), the six codons in the L1 region of *HTZ1* (the gene that encodes yeast H2A.Z) were individually mutated to a cysteine codon (*Figure 1B*). The resulting *HTZ1(Cys)* alleles were fused in-frame to a C-terminal 2xFLAG (indicated as FL) tag in a fragment containing *URA3*. These fragments were used to replace the endogenous *HTZ1* by homologous recombination. All six *HTZ1(Cys)*$^{FL}$ alleles fully complemented the formamide sensitivity of the *htz1Δ* mutant (*Figure 1C*) (*Wu et al., 2005*). Since wild-type core histones of yeast do not contain any cysteine, the Htz1(Cys)$^{FL}$ protein contributes the only sulfhydryl group to the nucleosome. The gene symbol of *HTZ1(Cys)*$^{FL}$ is capitalized because the crosslinking phenotype is dominant (see below).

The reducing environment within the yeast nucleus is prohibitive for cystine linkages to accumulate (*Dardalhon et al., 2012*). Intra-nucleosomal crosslinking of Htz1(Cys)$^{FL}$ was facilitated by the cell-permeable, thiol-specific oxidizing agent 4,4'-dipyridyl disulfide (4-DPS) (*Figure 1A*). To monitor the intra-nucleosomal crosslinking of Htz1(Cys)$^{FL}$, logarithmic yeast cultures were treated with 4-DPS or dimethylsulfoxide (DMSO, the negative control, indicated as "−" in *Figure 1D–E*) for 20 min before fixation and lysis in trichloroacetic acid (TCA). The precipitated histones were extracted for 60 min using a buffer called TUNES, which contains Tris base, urea, NaCl, EDTA, sodium dodecyl sulfate (SDS) and glycerol. To ensure that the disulfide linkages formed are representative of the protein-protein interactions captured *in vivo*, N-ethyl maleimide (NEM) was added to block any free thiol groups from crosslinking after cell lysis. Proteins were resolved by non-reducing SDS-polyacrylamide gel electrophoresis (SDS-PAGE) and analyzed by anti-FLAG immunoblotting. In the presence of 4-DPS, the H2A.Z-to-H2A.Z crosslinking adducts (ZxZ), which migrate at ~37 kD, were detected for cysteines substituted at A45, T46, G47 and R48, but not H44 or T49 (*Figure 1D–E*, *top panels*). The disulfide crosslinking efficiency, which is defined as the ratio of ZxZ over total H2A.Z immunoblotting signals, does not correlate particularly well with Cα-Cα' proximity, suggesting that the L1-L1' interface is dynamic (*Figure 1F* and *Figure 1—figure supplement 2*). The formation of the ZxZ adduct for A45C, T46C, G47C, and R48C was accompanied by a reduction in the FLAG-tagged Htz1 (Cys)$^{FL}$ monomer (mono Z) at ~17 kD (*Figure 1D–F*). Under reducing SDS-PAGE conditions where β-mercaptoethanol (β-ME) was added to the gel loading buffer, all ZxZ adducts were cleaved and the

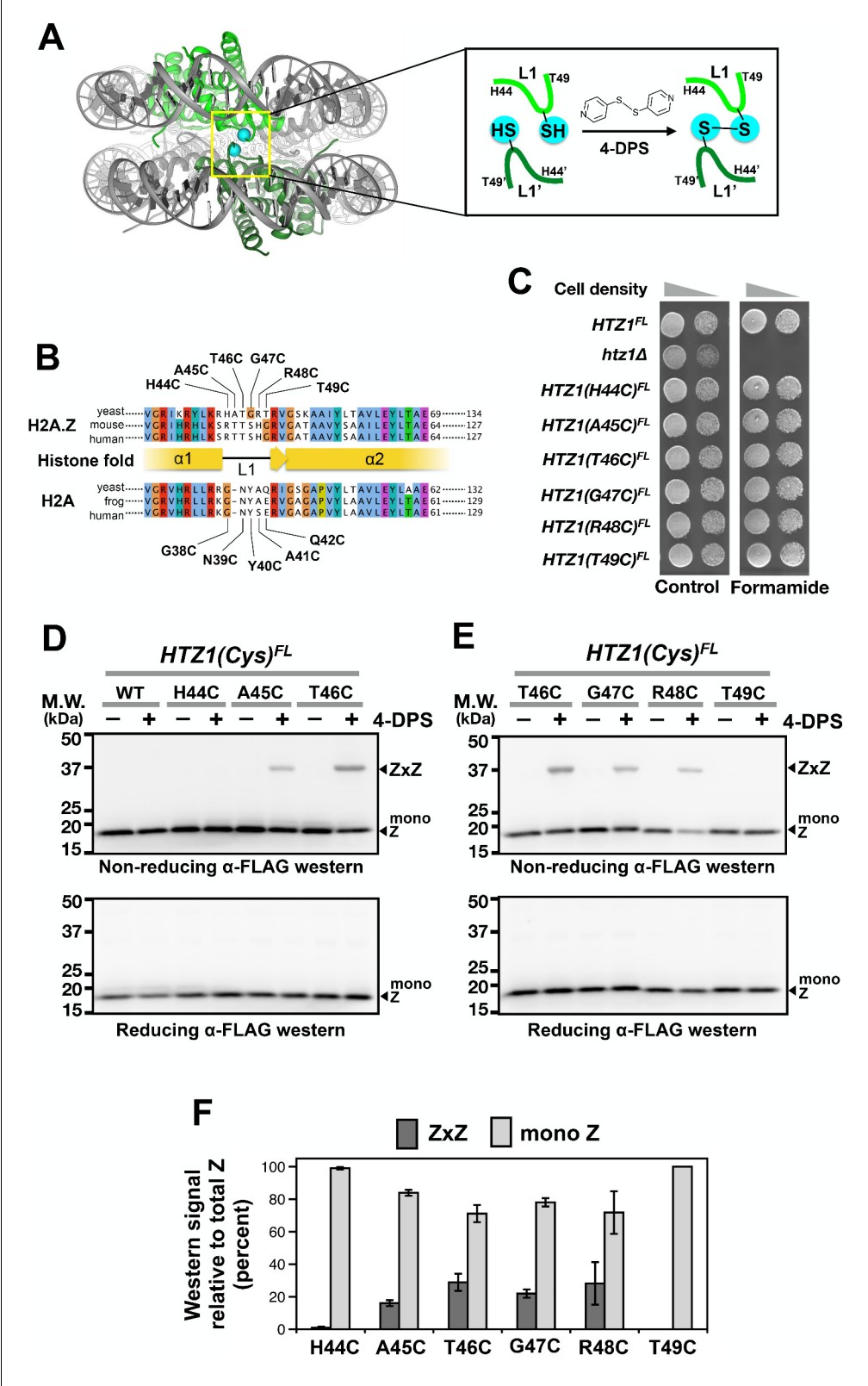

**Figure 1.** Cysteine substitution at multiple positions of the L1 region of H2A.Z supports Z-to-Z crosslinking in yeast cells. (**A**) The L1-L1' interface of the ZZ nucleosome (PDB: 1F66) is highlighted by a yellow box (**Suto et al., 2000**). The two nucleosomal H2A.Z-H2B dimers are highlighted in green. Cyan spheres mark the alpha-carbon (Cα) of T41 of mouse H2A.Z, which corresponds to the T46 position of yeast Htz1. Inset depicts the 4-DPS-dependent crosslinking reaction between the L1 cysteines of the two H2A.Z molecules within the ZZ nucleosome. Numbering of the amino acids refers to the yeast

*Figure 1 continued*

Htz1. (B) Sequence alignment analysis in and around the L1 region of H2A.Z and H2A from three different species was performed using the Clustal Omega algorithm in Jalview (*Waterhouse et al., 2009*). (C) Genetic complementation was performed by spotting equal number of cells (and 10-fold dilution in spots on right) of the indicated strains onto YPD media with and without 2.5% formamide. (D–E) VivosX analysis of yeast H2A.Z. Wild-type and *HTZ1(Cys)$^{FL}$* mutants that were treated with 180 µM of 4-DPS (+) or with DMSO (–) were analyzed by non-reducing (*top panel*) or reducing (*bottom panel*) SDS-PAGE followed by anti-FLAG immunoblotting. (F) Quantification of Z-to-Z crosslinking efficiency. Bars in *dark gray* (ZxZ) represent the mean ratios of ZxZ signal over total H2A.Z signal (i.e. ZxZ plus mono Z) in (D) and (E). Bars in light gray (mono Z) represent the mean ratios of mono Z over total H2A.Z. Means and standard deviations (error bars) were calculated from at least three biological replicates. *HTZ1(T46C)* were performed six times. ZxZ: Z-to-Z cystine adducts. Mono Z: uncrosslinked H2A.Z.

DOI: https://doi.org/10.7554/eLife.36654.002

The following source data and figure supplements are available for figure 1:

**Source data 1.** Values used to plot *Figure 1F*.
DOI: https://doi.org/10.7554/eLife.36654.007
**Figure supplement 1.** A cartoon depicting the proposed histone cycle.
DOI: https://doi.org/10.7554/eLife.36654.003
**Figure supplement 2.** Relative Cα-Cα' distances at the L1-L1' interface of AA and ZZ nucleosomes.
DOI: https://doi.org/10.7554/eLife.36654.004
**Figure supplement 3.** Effects of NEM blocking and cystine linkage stability.
DOI: https://doi.org/10.7554/eLife.36654.005
**Figure supplement 4.** The thiol-disulfide interchange reaction between the cysteine thiols and 4-DPS.
DOI: https://doi.org/10.7554/eLife.36654.006

Htz1(Cys)$^{FL}$ mutant proteins migrated as monomers (*Figure 1D–E*, bottom panels). The *HTZ1 (T46C)$^{FL}$* mutant was chosen for subsequent VivosX experiments because it also efficiently cross-linked to an L1' cysteine mutant in H2A (see below).

The effectiveness of the NEM blocking step was confirmed by the inhibition of total protein labeling with the thiol-specific fluorescence reagent, Alexa647-maleimide (*Figure 1—figure supplement 3A*, compare Lanes 1 and 3) and by the inhibition of formation of non-specific disulfide crosslinking adducts of Htz1(T46C)$^{FL}$ (*Figure 1—figure supplement 3B*, compare Lanes 1 and 3). The duration for the extraction of precipitated histones by the TUNES buffer was empirically determined to be 60 min as it allows maximal solubilization of the histones without any appreciable reduction of the disulfide adducts (*Figure 1—figure supplement 3C–D*).

To determine whether VivosX can detect H2A-H2A (AxA) crosslinks, a cysteine codon was substituted for various amino acids in the L1 loop of *HTA1* (one of the two paralogs that encode yeast H2A) in an episomal *[$^{V5}$HTA1-HTB1-HIS3]* vector (*Hirschhorn et al., 1995*). Although the mutated *HTA1(Cys)* alleles contain an N-terminal fusion to the V5 epitope, all immunoblots below were conducted using a pan-H2A antibody (Active Motif) since this antibody gave a stronger signal than the V5 antibody (not shown). Note that the L1 loop of H2A is one amino acid shorter than that of Htz1 (*Figure 1B*). To test for function, the five *[$^{V5}$HTA1(Cys)-HTB1-HIS3]* plasmids were transformed into an (*hta1-htb1)Δ (hta2-htb2)Δ* strain kept alive by a replicating plasmid carrying *[HTA1-HTB1-URA3]* (*Figure 2A*). Loss of the *URA3* plasmid was selected on medium containing 5-fluoroorotic acid (5-FOA) and was lethal (*Boeke et al., 1987*). This inviability was rescued by each one of the *$^{V5}$HTA1 (Cys)* mutants carried on the *HIS3* plasmid, indicating that H2A function was not impaired by the cysteine substitutions (*Figure 2B*).

Using the VivosX method described above but probing with anti-H2A antibodies, AxA crosslink adducts were observed for the H2A N39C, Y40C, A41C, and Q42C mutant proteins, but not G38C (*Figure 2C–D*). While the lack of G38C crosslinking can be explained by a long Cα-Cα' distance (13.2 Å), the crosslinking efficiency of the other four H2A L1 loop sites does not correlate well with Cα-Cα' proximity (*Figure 2E* and *Figure 1—figure supplement 2C*) (*White et al., 2001*). In fact, A41C and Q42C exhibited robust crosslinking despite having a Cα-Cα' distance of >10 Å. Similar to Htz1(Cys)$^{FL}$, an increase in AxA adducts was accompanied by a reciprocal decrease in monomeric H2A (mono A) in a 4-DPS-dependent manner (*Figure 2C–D*, top panels and *Figure 2E*). The AxA species was lost when the proteins were exposed to β-ME, further confirming that AxA is caused by disulfide crosslinking at the cysteine sites (*Figure 2C–D*, bottom panel). Note that low levels of AxA

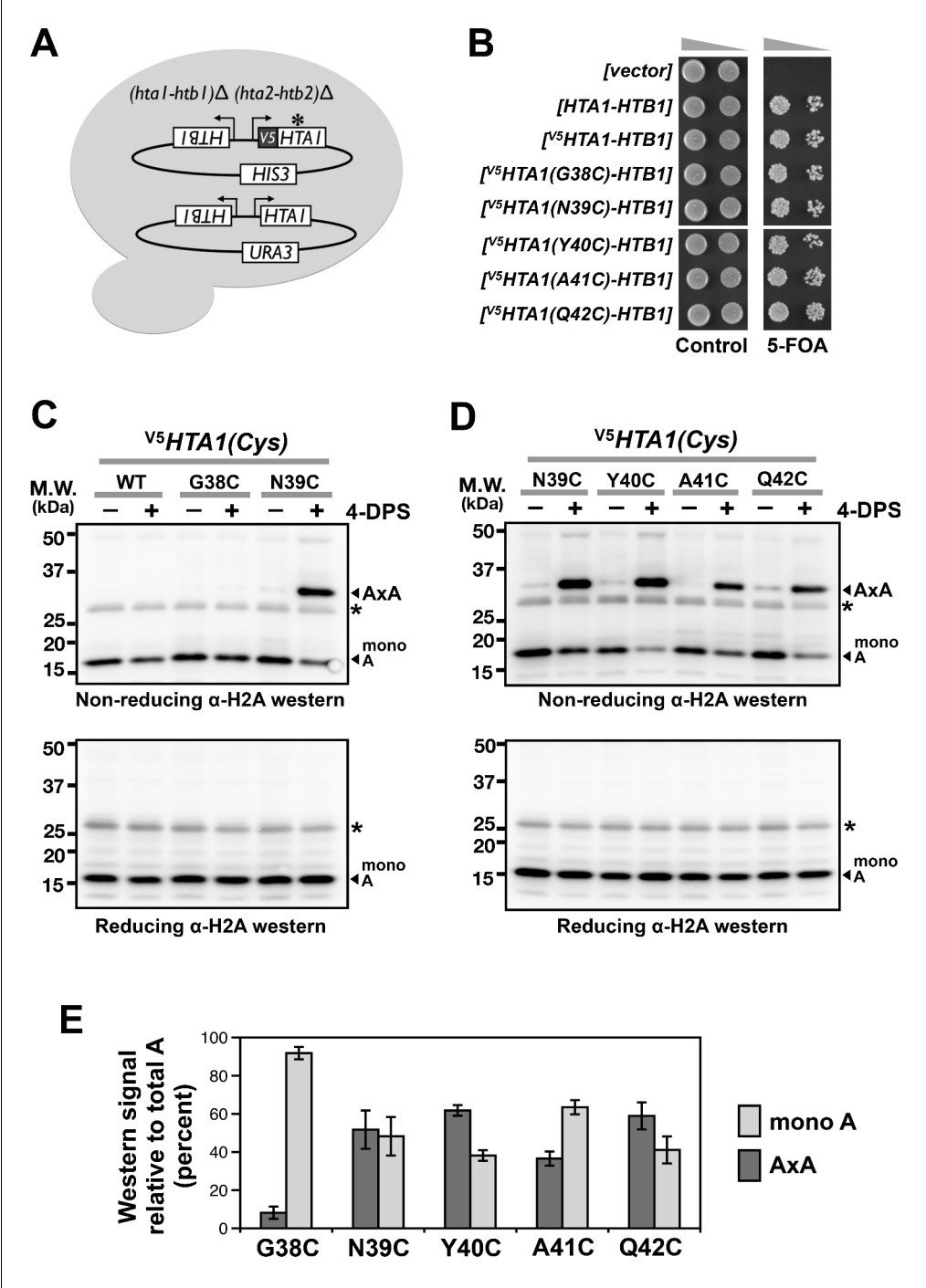

**Figure 2.** Cysteine substitution at multiple positions of the L1 region of H2A supports A-to-A crosslinking in yeast cells. (A) The cartoon depicts the plasmid shuffle yeast system used to verify the functionality of the $^{V5}HTA1(Cys)$ mutants. (B) The ability of the *HTA1(Cys)* mutants to complement the lack of endogenous genes for H2A and H2B was indicated by growth in the presence of 5-FOA, which removes the wild type [*HTA1-HTB1-URA3*] vector from the cells. (C,D) VivosX analysis of yeast H2A was performed as described in *Figure 1D* except that the immunoblots were probed with an anti-H2A antibody (*Active Motif*). Asterisk (*) indicates a non-specific band. (E) Quantification of the AxA adducts and the uncrosslinked H2A (mono A) was performed as described in *Figure 1F*. Bars and error bars indicate the means and standard deviation of three biological replicates.

DOI: https://doi.org/10.7554/eLife.36654.008

The following source data is available for figure 2:

**Source data 1.** Values used to plot *Figure 2E*.
DOI: https://doi.org/10.7554/eLife.36654.009

crosslinks were detected for N39C, Y40C, A41C and Q42C even in the absence of 4-DPS. This may reflect the natural redox state of the nucleus (*Dardalhon et al., 2012*).

VivosX was also used to probe heterotypic AZ nucleosomes using H2A.Z-to-H2A crosslinking. Cells expressing $^{V5}HTA1(N39C)$ or $^{V5}HTA1$ were combined with $HTZ1(T46C)^{FL}$ or $HTZ1^{FL}$ as the sole source of these histones. After treating the cells with 4-DPS, total histones were extracted and examined by non-reducing SDS-PAGE and immunoblotting as described above. As before, strains containing $HTZ1(T46C)^{FL}$ exhibited a 4-DPS-dependent ZxZ adduct (*Figure 3A*, lanes 6 and 8), while those containing $^{V5}HTA1(N39C)$ exhibited an AxA adduct (*Figure 3B*, lanes 4 and 6). For the $^{V5}HTA1(N39C)\ HTZ1(T46C)^{FL}$ double mutant, two crosslinked species were observed when the blots were probed with either anti-FLAG or anti-H2A to detect H2A.Z and H2A, respectively (*Figure 3A*, lane 6 and *Figure 3B*, lane 6). In each case, one band of the doublet co-migrated with the homotypic adduct. For H2A.Z, the top band exhibited the same mobility as ZxZ (*Figure 3A*, lanes 6 and 8), while the lower band of the H2A doublet co-migrated with AxA. The differences in ZxZ and AxA mobility are consistent with the observation that monomeric Htz1(T46C)$^{FL}$ migrates slower than monomeric $^{V5}$Hta1(N39C) despite the fact that the calculated molecular weight of Htz1(T46C)$^{FL}$ is actually smaller (16.6 kD versus 16.8 kD), possibly due to the strong negative charge of the FLAG epitope (*Figure 3A–B*, *Figure 3—figure supplement 1A–B*). A doublet was observed only when $^{V5}HTA1(N39C)$ and $HTZ1(T46C)^{FL}$ were combined, suggesting that one band of the doublet represents $^{V5}$Hta1(N39C) crosslinked to Htz1(T46C)$^{FL}$ (AxZ) (*Figure 3A–B*, lane 6).

To determine whether the extra band of the doublet is indeed the AxZ adduct, two aliquots of the $^{V5}HTA1(N39C)\ HTZ1(T46C)^{FL}$ extract were loaded side-by-side in a non-reducing SDS polyacrylamide gel separated only by a lane with molecular weight markers (*Figure 3C*). The immunoblot was cut in the middle of marker lane and the two halves were probed separately with anti-FLAG and anti-H2A antibodies. Realignment of the two halves of the blot demonstrated that the predicted AxZ band contains both $^{V5}$Hta1 and Htz1$^{FL}$ and runs between the ZxZ and AxA adducts (*Figure 3C*, *top panel*). All the crosslinked proteins were eliminated when fractionated on a gel in the presence of β-ME, confirming that, similar to ZxZ and AxA, the AxZ adduct was also generated by disulfide crosslinking (*Figure 3C*, *bottom panel*).

An advantage of VivosX is that it uses whole cell extracts, thereby obviating the need to purify chromatin, which greatly facilitates high-throughput experiments. The assumption is that only histones in close proximity in assembled nucleosomes will crosslink and therefore the AxA, AxZ, and ZxZ signals derived from whole cell extracts accurately reflect the native AA, AZ and ZZ nucleosomal levels. If true, then similar ratios of the various crosslinked species should be observed when native nucleosomes are extracted from cells before they are used for crosslinking. This assumption was tested by isolating chromatin from the $^{V5}HTA1(N39C)$ and/or the $HTZ1(T46C)^{FL}$ strains and digesting it with micrococcal nuclease (MNase) to generate a pool of soluble native nucleosomes (*Figure 3—figure supplement 1C*). These nucleosomes were then incubated with 4-DPS or DMSO for 20 min followed by non-reducing SDS-PAGE analysis and the immunoblots probed as before with either anti-FLAG or anti-H2A antibodies (*Figure 3D*). The ratios of AxZ to ZxZ (1:0.2) or AxZ to AxA (1:3.5) in native nucleosomes (*Figure 3D*, lanes 3 and 5) were similar to those observed with the $^{V5}HTA1(N39C)\ HTZ1(T46C)^{FL}$ double mutant using total extracts (1:0.3 and 1:4.2, respectively) (*Figure 3A–C*, averaged values).

Higher levels of the AxA, AxZ, and ZxZ adducts were present in native nucleosomes than whole cell extracts of the untreated cells (*Figure 3D*, 4-DPS minus lanes). This difference was likely due to atmospheric oxidation of the dithiols during the cell lysis, chromatin preparation and/or MNase digestion steps. Such background oxidation was minimized in the whole cell extracts as cells were fixed by TCA and extraction was performed in the presence of the thiol blocker NEM. TCA and NEM were added to the native nucleosomes only after MNase digestion. Note also that there is an extra band between the 37 and 50 kD markers in the anti-H2A blot of native nucleosomes. The nature of this H2A adduct is currently unknown.

## VivosX accurately reports changes in global H2A.Z occupancy in various mutants

An important question is whether the intra-nucleosome interactions captured by VivosX can be used as a proxy to report on global *in vivo* H2A.Z occupancy. If this is the case, then mutants that either increase or decrease H2A.Z levels on chromosomes, should exhibit similar changes in ZxZ, AxZ and

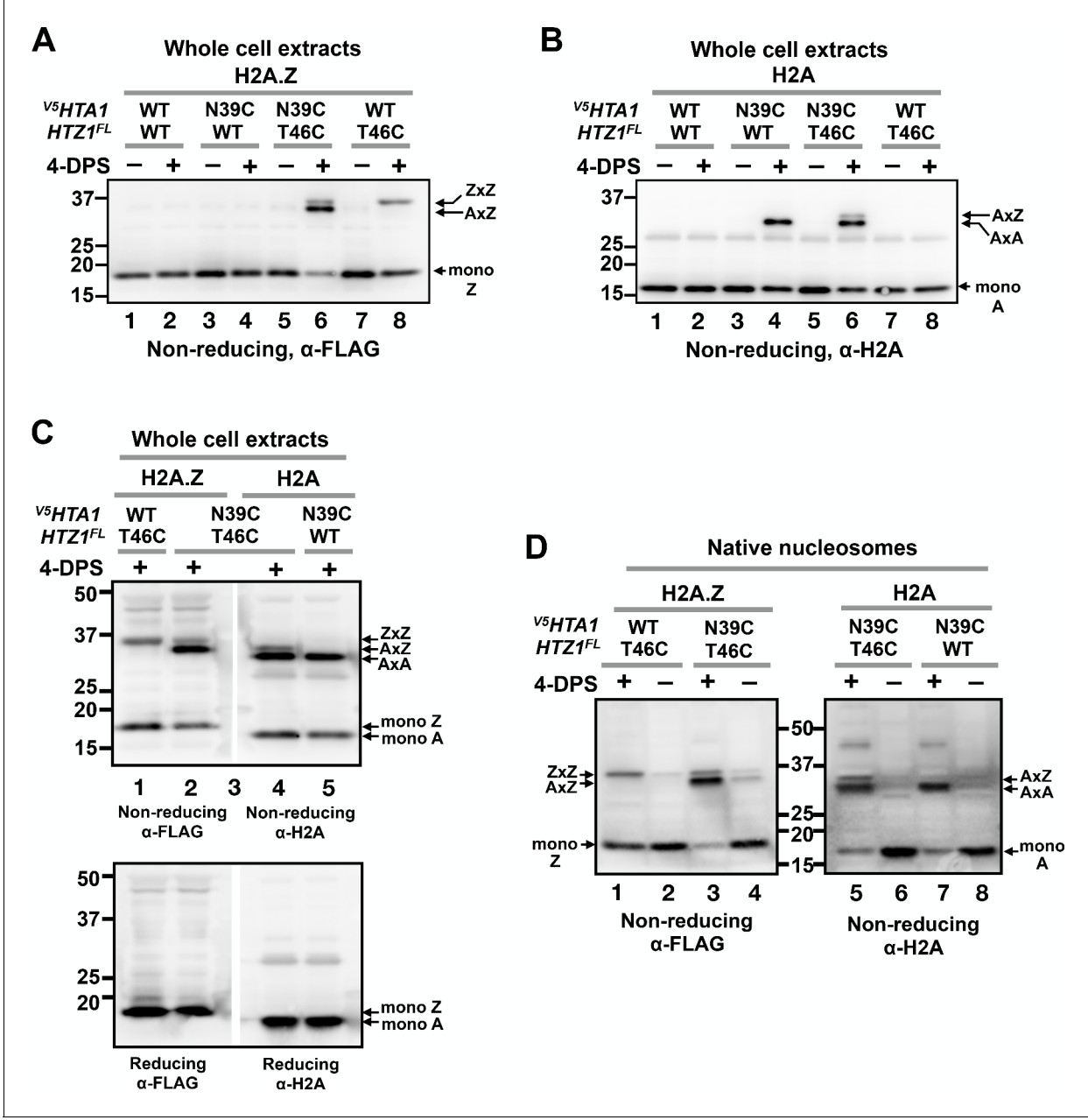

**Figure 3.** VivosX distinguishes AA, AZ and ZZ nucleosomal species. (**A**) Yeast cells expressing $^{V5}HTA1(N39C)$ or $^{V5}HTA1$ in combination with $HTZ1$ $(T46C)^{FL}$ or $HTZ1^{FL}$ were analyzed by VivosX as described in *Figure 1D*. An anti-FLAG antibody was used to detect the FLAG-tagged H2A.Z and its crosslink adducts. (**B**) Same as (**A**), except that the immunoblot was probed with anti-H2A antibody. (**C**) The indicated samples were analyzed as in (**A**) and (**B**). But after the proteins were transferred to a PVDF membrane, the membrane was cut in the middle of lane 3, which contained the molecular weight markers. The two halves of the membranes were probed with either the anti-FLAG or anti-H2A antibodies. The *top* and *bottom panels* were analyzed by non-reducing and reducing SDS-PAGE, respectively. (**D**) Native nucleosomes released by MNase digestion of the chromatin pellets prepared from the indicated strains were incubated with 4-DPS (+) or DMSO (-). After TCA precipitation and extraction with the TUNES buffer, the histones were analyzed by non-reducing SDS-PAGE and anti-FLAG (*left*) or anti-H2A (*right*) immunoblotting.

DOI: https://doi.org/10.7554/eLife.36654.010

The following figure supplement is available for figure 3:

**Figure supplement 1.** Control experiments for yeast VivosX.

DOI: https://doi.org/10.7554/eLife.36654.011

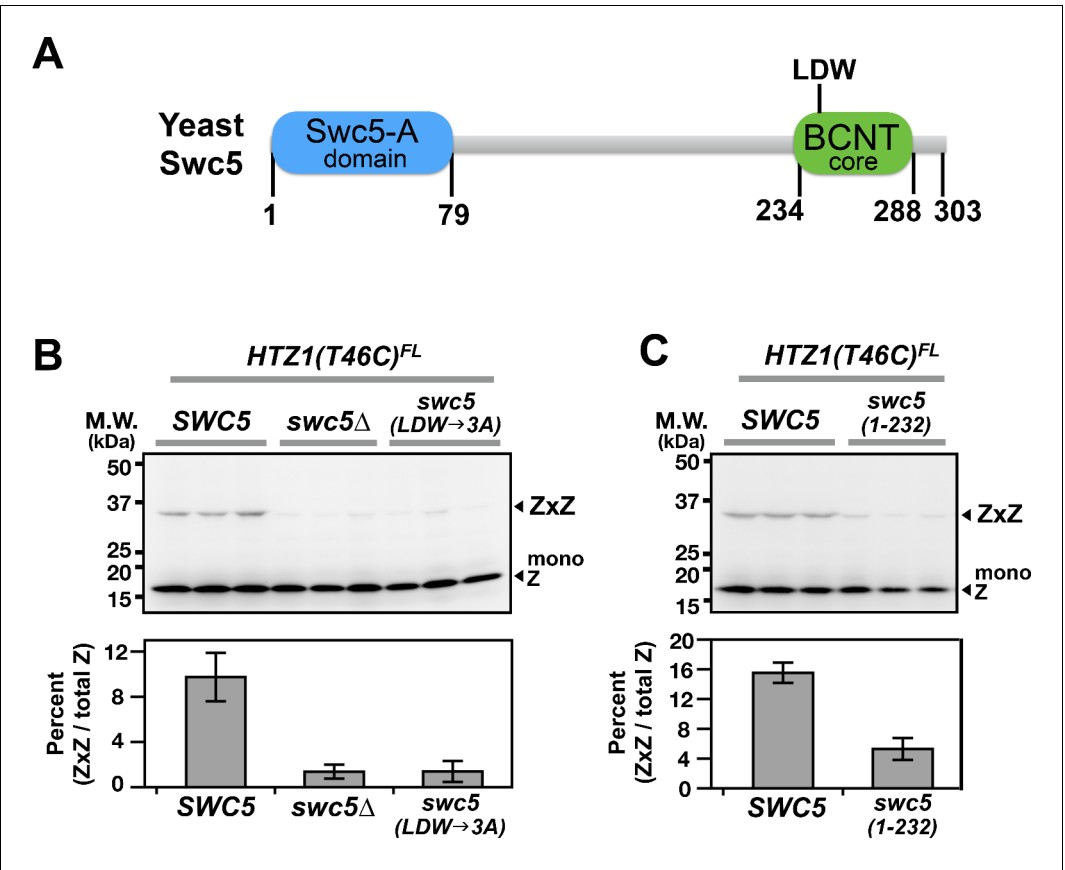

**Figure 4.** Domain analysis of Swc5 using H2A.Z VivosX. (**A**) The cartoon depicts the domain organization of yeast Swc5 (**Sun and Luk, 2017**). (**B,C**) The yeast strain, *HTZ1(T46C)*[FL] *swc5Δ*, was transformed by a single-copy plasmid containing either the wild-type *SWC5* or the indicated *SWC5* mutants or by the control vector (*swc5Δ*). *Top panels:* Each strain was represented by three independent transformants and analyzed by VivosX in parallel. Htz1(T46C)[FL] and its crosslinked (ZxZ) adducts was detected by anti-FLAG immunoblotting. *Bottom panels:* Quantification of the immunoblots above. Bars and error bars represent the means and standard deviations of three biological replicates.

DOI: https://doi.org/10.7554/eLife.36654.012

AxA adducts using VivosX. *SWC5* encodes a component of the SWR complex that is required for its H2A.Z deposition activity at promoters (**Tramantano et al., 2016**). In the absence of *SWC5*, H2A.Z levels at promoters are decreased (**Tramantano et al., 2016**). A *swc5Δ HTZ1(T46C)*[FL] haploid was transformed with either a vector plasmid or one carrying *SWC5*. VivosX was performed on these strains as described above. ZxZ adducts were observed in the extracts from the *SWC5* strain, but not *swc5Δ* (**Figure 4B**). Additional mutants of *SWC5* that were previously shown to prevent H2A.Z deposition include a truncation of the highly conserved BCNT domain [*swc5(1-232)*] and substitution of the LDW motif within BCNT with alanines [*swc5(LDW→3A)*] (**Sun and Luk, 2017**) (**Figure 4A**). These mutants also significantly reduced ZxZ crosslinking efficiency in VivosX (**Figure 4B–C**). Note that *swc5Δ HTZ1(T46C)*[FL] carrying *SWC5* on a plasmid has a lower ZxZ crosslinking efficiency than *HTZ1(T46C)*[FL] with endogenous *SWC5* (compare **Figure 4** to **Figure 1F**). The discrepancy could be due to loss of the *SWC5* plasmid in a fraction of cells during growth of the culture.

VivosX detected a decrease in global ZZ nucleosome occupancy when SWR activity was compromised. Can it also detect increases in ZZ occupancy when ZZ nucleosome disassembly is impaired? ZZ nucleosomes are normally removed from the +1 position at promoters by formation of the transcriptional preinitiation complex (PIC) (**Tramantano et al., 2016**). As a result, ZZ nucleosomes accumulate when their disassembly is inhibited by depletion of the TATA-binding protein (TBP), which is required for PIC recruitment (**Hahn, 2004**). TBP depletion was effected using a fusion protein of TBP

and the FKBP12-rapamycin-binding domain (FRB). This fusion is dragged out of the nucleus by binding to the FKBP12 tag on the pre-ribosome in a rapamycin-dependent manner (*Haruki et al., 2008*). To test if VivosX can detect a conditional block in H2A.Z eviction, *HTZ1(T46C)*[FL] was introduced into the *TBP-FRB* Anchor-away strain or the untagged (no FRB) control (*Haruki et al., 2008*). After treating these cells with rapamycin for 1 hr to block PIC assembly and H2A.Z eviction, 4-DPS was added to induce ZxZ crosslinking. A reproducible increase of ZxZ adducts and a concomitant decrease of mono Z was observed after TBP-FRB depletion ($N = 3$), consistent with our earlier quantitative ChIP-seq results (*Figure 5A*, compare lanes 6 and 8) (*Tramantano et al., 2016*). By contrast, ZxZ crosslinks were reduced when *SWC5-FRB* was depleted by Anchor-away ($N = 3$) (*Figure 5A*, lanes 10 and 12). VivosX can therefore accurately report the global chromatin dynamics of H2A.Z.

## ZZ nucleosomes are preferentially disassembled by the PIC

A block in the assembly of the PIC leads to H2A.Z accumulation at promoters, but whether the presence of a single H2A.Z (i.e. AZ) is sufficient for PIC-mediated nucleosome disassembly, or whether only ZZ nucleosomes are removed was unknown (*Tramantano et al., 2016*) (*Figure 1—figure supplement 1*). Because VivosX can detect both AZ and ZZ nucleosomes, it could be used to answer this question. The [[V5]*HTA1(N39C)-HTB1-HIS3*] plasmid was introduced into the *HTZ1(T46C)*[FL] *TBP-FRB* Anchor-away strain. Note that the endogenous genes for H2A were present in this strain and therefore not all AA and AZ nucleosomes will be crosslinked, contrasting the double mutant used in *Figure 3*. Rapamycin was added to half of the culture to block PIC-dependent H2A.Z eviction by depleting *TBP-FRB* from the nucleus. The cultures with and without rapamycin were then treated with 4-DPS, fixed by TCA and analyzed by SDS-PAGE and immunoblotting with anti-FLAG antibodies. In the presence of *TBP-FRB* both ZxZ and AxZ crosslinks were detected at ~37 kD, (*Figure 5B*). Blocking PIC assembly by depleting TBP-FRB preferentially increased the ZxZ adduct, indicating that ZZ nucleosomes, but not AZ nucleosomes, were targeted by PIC for removal (*Figure 5B–C*). In fact, monomeric H2A.Z is diminished reciprocally, consistent with the free pool of Z-B dimers being used by SWR to generate more ZZ nucleosomes, but without being replenished by PIC-dependent ZZ nucleosome disassembly (*Figure 1—figure supplement 1*). While the increase in the ZxZ adduct and the corresponding decrease in monomeric H2A.Z were small, these differences were reproducible in three biological replicates and statistically significant, with non-overlapping standard deviations and p values of $< 0.05$ based on t-tests. This experiment illustrates the power of the VivosX assay for studying H2A.Z dynamics.

## VivosX can detect intra-nucleosomal interactions in human cells

Histone proteins are highly conserved, raising the possibility that VivosX could be applied to probe intra-nucleosomal interactions in other species besides yeast. VivosX was therefore used to examine ZZ nucleosomes in human cells. Human H2A.Z has two isoforms (H2A.Z.1 and H2A.Z.2), and they are encoded by the *H2AFZ* and *H2AFV* genes, respectively (*Vardabasso et al., 2014*). VivosX was applied to *H2AFZ* by substituting a cysteine for the L1 residue H43 of H2A.Z.1 (referred to as H2A.Z hereafter) (*Figure 1B*). A V5 tag was fused to the C-terminus of H2A.Z to facilitate detection. H2A.Z (H43C) corresponds to the yeast Htz1(R48C), which has a similar crosslinking efficiency as the Htz1 (T46C) used in most yeast experiments (*Figure 1B and F*). H2A.Z(H43C) was chosen for human VivosX, instead of H2A.Z(T41C) [cognate amino acid of yeast (T46C)] because the crosslinking efficiency of Htz1(R48C) was slightly better than Htz1(T46C) in our preliminary experiments (data not shown).

Lentiviral vectors bearing a tetracycline-inducible *H2AFZ*[V5] or *H2AFZ(H43C)*[V5] alleles were used to infect the non-transformed human mammary epithelial cell line, MCF10A. The transduced cells were selected to establish stable cell lines, which constitutively express a reverse tetracycline transactivator to allow inducible ectopic expression of *H2AFZ(H43C)*[V5] or *H2AFZ*[V5] in the presence of doxycycline (*Leung and Brugge, 2012*). The cells were then incubated with 4-DPS or DMSO for 20 min and lysed with TUNES buffer in the presence of NEM. Total cell lysates were then analyzed by non-reducing or reducing SDS-PAGE and immunoblots were probed with anti-V5 antibodies.

The H2A.Z(H43C)[V5] protein exhibited a more complex pattern of crosslinked proteins than yeast Htz1(T46C)[FL] (*Figure 6A*, lane 4). Four slower migrating bands appeared above the expected monomeric H2A.Z(H43C)[V5]. The top three bands in H2A.Z(H43C)[V5] lane were disulfide adducts that occur at the H43C position because these bands were not observed in the cells expressing wild-type H2A.

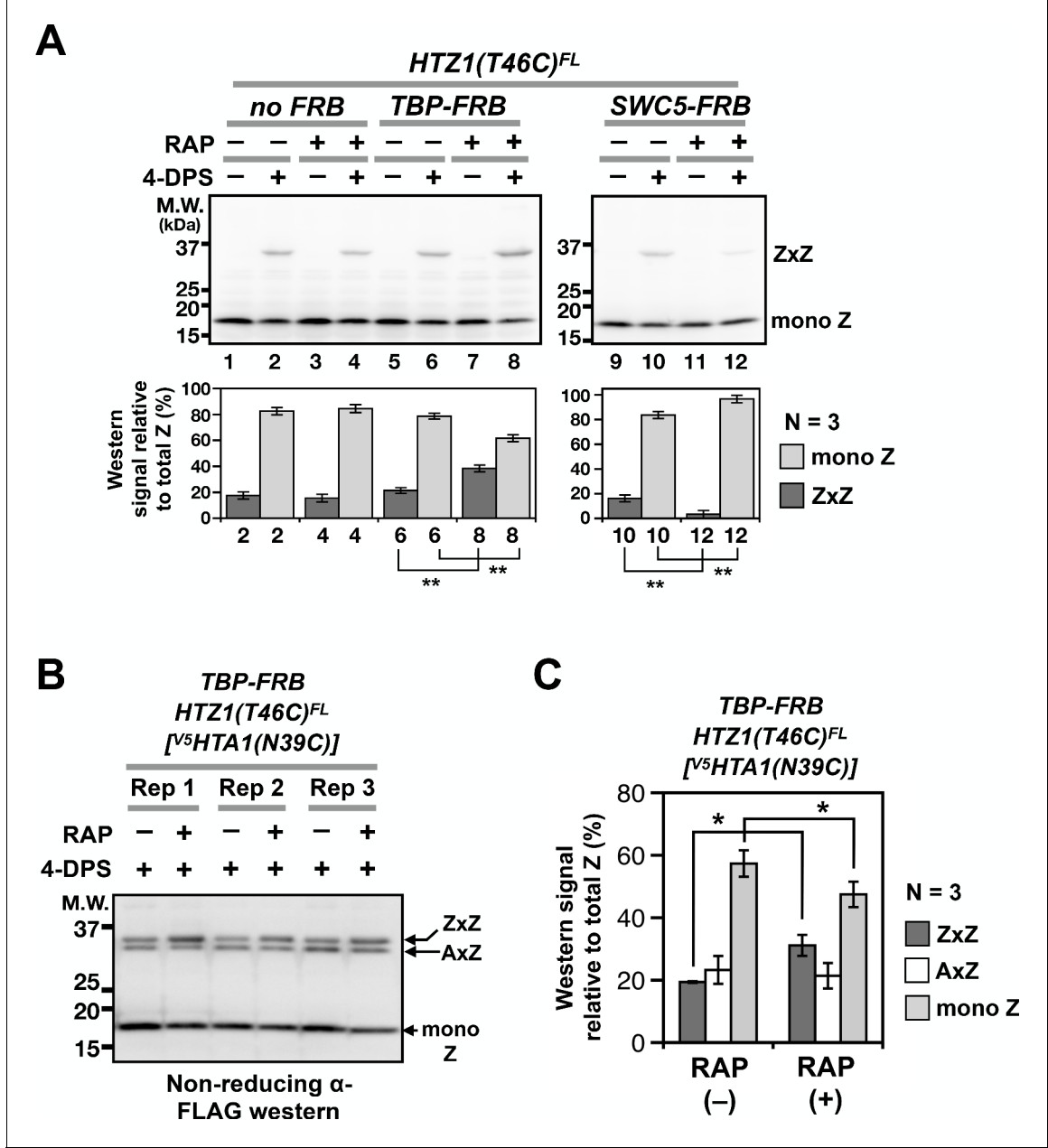

**Figure 5.** Probing H2A.Z dynamics using VivosX. (**A**) *HTZ1(T46C)*[FL] strains containing the *TBP-FRB* or *SWC5-FRB* alleles or the corresponding wild-type alleles (*no FRB*) in the Anchor-away genetic background (W303 MAT a *tor1-1 fpr1 RPL13A-2xFKBP12*) (*Haruki et al., 2008*) were incubated with rapamycin (+RAP) or without (i.e. DMSO, –RAP) for 1 hr before each culture was divided into halves, where one half was oxidized with 4-DPS (for 20 min) and the other without. Fixation, protein extraction, and immunoblotting analysis were conducted as described for *Figure 1D*. *Bottom panels*: Quantification of the ZxZ bands and the mono Z bands was performed as described in *Figure 1D*. (**B**) VivosX was performed using *HTZ1(T46C)*[FL] *TBP-FRB* yeast transformed with the [[V5]*HTA1(N39C)-HTB1*] plasmid. Rep: biological replicates. (**C**) Quantification of (**B**). The immunoblot signals of ZxZ, AxZ, and mono Z were normalized to total H2A.Z. Bars and error bars represent the means and standard deviations of three biological replicates. One asterisk (*) indicates p≤0.05 and two (**) indicates p≤0.01 of t-tests.

DOI: https://doi.org/10.7554/eLife.36654.013

The following source data is available for figure 5:

**Source data 1.** Values used to plot *Figure 5A*.
DOI: https://doi.org/10.7554/eLife.36654.014

**Source data 2.** Values used to plot *Figure 5C*.
DOI: https://doi.org/10.7554/eLife.36654.015

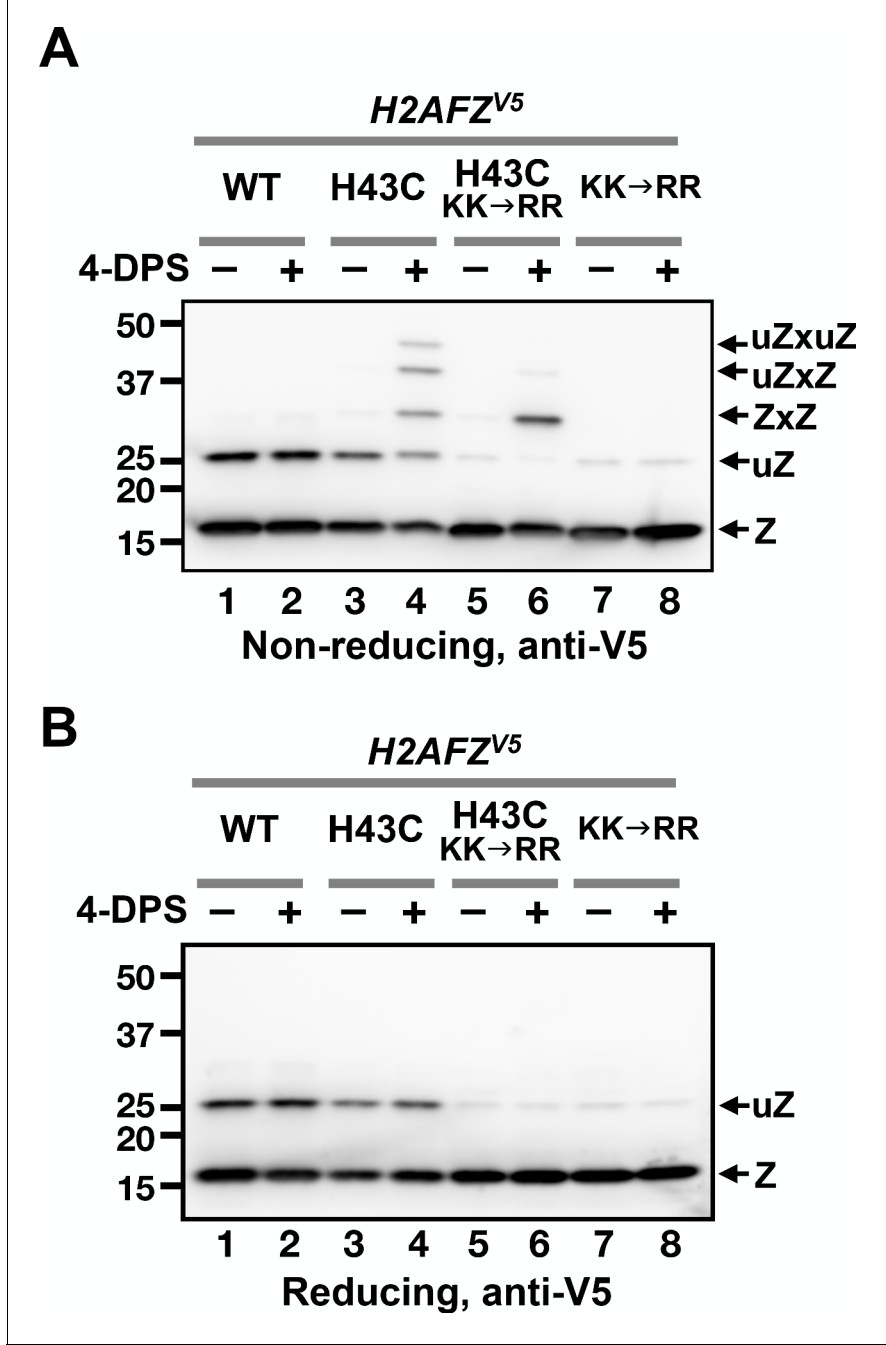

**Figure 6.** VivosX of H2A.Z in human cells. (A, B) Human MCF10A cells expressing an ectopic *H2AFZ(H43C)*[V5] gene was incubated with 4-DPS for 20 min before lysis using the TUNES buffer. Total lysates were resolved by non-reducing SDS-PAGE in (A) and reducing SDS-PAGE in (B) before analyzed by anti-V5 immunoblotting. Z: nonubiquitylated H2A.Z; uZ: monoubiquitylated H2A.Z; ZxZ: Z-to-Z crosslink adducts; uZxZ: crosslink adducts with one uZ and one Z; uZxuZ: crosslink adducts of two uZ molecules.

DOI: https://doi.org/10.7554/eLife.36654.016

The following figure supplements are available for figure 6:

**Figure supplement 1.** No apparent H3-to-H3 crosslinking observed after 4-DPS treatment.
DOI: https://doi.org/10.7554/eLife.36654.017
**Figure supplement 2.** Estimation of ectopic H2A.Z level.
DOI: https://doi.org/10.7554/eLife.36654.018

Z$^{V5}$ (*Figure 6A*, compare lanes 2 and 4). In addition, when the same lysate of H2A.Z(H43C)$^{V5}$ was treated with β-ME, these three bands disappeared (compare Lane four in *Figure 6A–B*).

Human H2A.Z is monoubiquitylated and this modification is mainly associated with the H2A.Z found in facultative heterochromatin in human cells as opposed to the non-ubiquitylated form found predominantly in euchromatin (*Sarcinella et al., 2007*). Since the H2A.Z band near the 25 kD marker is not due to crosslinking, it could represent a ubiquitylated form of H2A.Z, as suggested by *Sarcinella et al. (2007)*. To test this idea, two (of three) ubiquitylation sites at K120 and K121 were mutated to arginines in the *H2AFZ$^{V5}$* and *H2AFZ(H43C)$^{V5}$* alleles. Both mutants reduced the amount of the ~25 kD band in the absence of 4-DPS, indicating that this band represents monoubiquitylated H2A.Z (*Figure 6A*, lanes 5–8). The two slowest crosslinked adducts were almost eliminated in the *H2AFZ(H43C,KK→RR)$^{V5}$* mutant, with a corresponding increase in the remaining adduct, suggesting that the ~30 kD band is the ZxZ disulfide adduct of two non-ubiquitylated H2A.Z monomers (*Figure 6A*, compare lanes 4, 6, and 8). Finally, the top two bands of the crosslinked H2A.Z(H43C)$^{V5}$ species are likely ZxZ adducts that are conjugated to one (uZxZ) or two ubiquitin moieties (uZxuZ). In H2A.Z(H43C, KK→RR)$^{V5}$, both of these upper bands were strongly diminished, indicating that these positions are associated with the ubiquitylated forms of H2A.Z (*Figure 6A*, compare lanes 4 and 6).

## Discussion

### VivosX is a powerful, cost-effective and efficient approach to assay intra-nucleosomal dynamics

The dynamics of histone core composition and conformation is an emerging theme in the understanding of chromatin remodeling activities and downstream genome functions. At +1 nucleosomes, for instance, the cycling of the AA, AZ and ZZ compositional states, is somehow linked to histone octamer eviction and efficient transcriptional response (*Luk et al., 2010*; *Tramantano et al., 2016*). VivosX revealed for the first time that the ZZ nucleosomes are preferentially disassembled by the transcription machinery. Traditionally to address such a question, biochemical fractionation of AZ and ZZ nucleosomes (before and after a block in PIC assembly) would be necessary. This approach involves technically non-trivial techniques such as sequential immunoprecipitation and relies on antibodies that may be limiting in quantity or expensive (*Luk et al., 2010*). In contrast, VivosX uses immunoblots of whole cell extracts, underscoring the simplicity and cost-effectiveness of the VivosX approach. The feasibility of extending VivosX to mutant screens for genes required for H2A.Z deposition and eviction makes the VivosX technique powerful.

One potential caveat of VivosX is the 'observer effect', where the crosslinking of two histones could potentially perturb *in vivo* chromatin dynamics. For example, Z-to-Z crosslinking might interfere with ZZ nucleosome disassembly. As such, VivosX may overestimate the *in vivo* level of ZZ nucleosomes. Alternatively but not exclusively, Z-to-A and A-to-A crosslinking could interfere with the H2A.Z deposition function of SWR as the remodeler replaces each of the two nucleosomal A-B dimers with Z-B dimers one at a time (*Luk et al., 2010*). Therefore, in the experiment where VivosX was used to predict changes in H2A.Z occupancy (*Figure 5*), H2A.Z eviction or deposition was blocked for at least 1 hr (a duration exceeding the turnover half-life of H2A.Z at a typical promoter) before the induction of crosslinking (*Tramantano et al., 2016*). Despite this precaution and the fact that H2A.Z VivosX correctly reported the change in H2A.Z occupancy, we cannot rule out that the L1-L1' disulfide crosslinking is indirectly interfering with H2A.Z deposition and eviction.

A second potential caveat was that 4-DPS, the oxidizing agent used in VivosX, could elicit an oxidative stress response as has been observed for other oxidizing agents (*Gasch et al., 2000*; *Weiner et al., 2012*). The transcriptional perturbation resulting from the oxidative stress could contribute to a change in H2A.Z occupancy and thus crosslinking efficiency. However, this does not appear to be the case. In a control experiment, sodium azide was added in conjunction with 4-DPS during the crosslinking step to inhibit ATP-dependent processes, such as transcription and replication. Similar levels of the ZxZ adducts were observed in the presence or absence of azide after 4-DPS treatment, suggesting that bulk Z-to-Z crosslinking is not dramatically affected by the oxidative stress response (*Figure 3—figure supplement 1D–E*), although rearrangement of the genomic ZZ nucleosome occupancy cannot be ruled out. Thus, interpretation of VivosX data must take the oxidative response caveat and the observer effect into consideration.

## Implications on nucleosomal dynamics

The average distance between the two alpha carbon atoms (Cα-Cα') of a disulfide bond is 5.6 Å with a range of 4.3 Å – 6.5 Å (*Schmidt et al., 2006*). It was therefore surprising that multiple positions along the L1 region of H2A and H2A.Z supported robust intra-nucleosomal crosslinking given that some of these positions have Cα-Cα' distances > 7 Å in crystal structures (*Figure 1—figure supplement 2C*) (*Suto et al., 2000*; *White et al., 2001*). Furthermore, some of the crosslinkable L1 side chains (e.g. Y40 and A41 of H2A) are largely inaccessible to solvent (*Suto et al., 2000*; *White et al., 2001*). Given that the thiol-disulfide interchange between cysteine thiols and 4-DPS involves the $S_N2$ nucleophilic substitution mechanism (*Figure 1—figure supplement 4A*), the L1 loop must adopt an alternative conformation from that observed in the crystal structure to allow the inaccessible cysteine thiolate to attack the disulfide of 4-DPS. We speculate that the nucleosomal L1-L1' interface between the two H2A.Z and/or H2A is dynamic. It has been proposed that the two halves of the nucleosome could transiently open like a clam, as they split between the two DNA gyres while the dyadic DNA region acts as a hinge (*Andrews and Luger, 2011*) (*Figure 1—figure supplement 4B*). Such a clamshell opening motion could further increase the degree of freedom of the L1-L1' interface, promoting VivosX adducts of H2A and H2A.Z.

In addition to the aforementioned solvent accessibility issue, the biochemical environment surrounding the cysteine residues can also influence disulfide crosslinking efficiency (*Singh and Whitesides, 1993*). The first step of the thiol-disulfide interchange reaction requires deprotonation of the cysteine sulfhydryl and thus will be facilitated by lowering of the p$K_a$ of the cysteine thiol (*Figure 1—figure supplement 4A*). The resulting thiolate then attacks the disulfide on 4-DPS to form a trisulfur anionic intermediate [$^{\delta\text{-}}$S–S–S$^{\delta\text{-}}$], which will be promoted by a basic environment and clearance from steric/positional constraints. Finally, the second $S_N2$ reaction will involve the opposite cysteine, again in thiolate form, to attack the mixed disulfide bond from the cysteine side to form a cystine, which is affected by the cysteine p$K_a$ and the position and orientation of the thiolate. Altogether, multiple factors could affect the crosslinking efficiency of L1-L1' at the different cysteines on L1, explaining why crosslinking efficiency does not simply correlate with Cα-Cα' distance of the cysteine pairs (*Figure 1—figure supplement 2C*).

A recent study from the Richmond lab indicated that recombinant *Xenopus* H2A with the corresponding cysteine substitution as yeast Hta1(N39C) and Htz1(T46C) supported efficient L1-L1' crosslinking of histone octamers *in vitro* (*Frouws et al., 2018*). Therefore, the majority of the nucleosomes (including the AA, AZ and ZZ configurations) in the *HTA1(N39C) HTZ1(T46C)* double mutant in *Figure 3* are expected to be crosslinked upon 4-DPS treatment. Notwithstanding a minor contribution of the uncrosslinked monomers a result of inefficient crosslinking, the bulk of the uncrosslinked species should represent the non-nucleosomal pools of A-B and Z-B dimers, which constitute 39% and 26% of total H2A and H2A.Z, respectively (*Figure 3A* lane 6, **3B** lane 6, and **3C** lanes 2 and 4). These unincorporated histones could be A-B and Z-B dimers that are associated with histone chaperones or chromatin remodelers, like SWR (*Luk et al., 2007*; *Sun and Luk, 2017*; *Wu et al., 2005*). These dimers could also be associated with chromatin in the form of hexasomes (partial nucleosomal particles missing one A-B or Z-B dimer) or non-nucleosomal histone-DNA complexes (*Andrews et al., 2010*; *Arimura et al., 2012*).

Although native yeast core histones are conveniently cysteine free, other non-histone proteins subjected to VivosX may not be. To determine how much background crosslinking is contributed by the endogenous cysteines on an interacting protein pair, 4-DPS treatment of cells expressing the unmodified genes should be included as negative controls. However, we argue that endogenous cysteines that are outside of a ~ 15 Å radius from the intended cysteine probe should not interfere with VivosX and thus need not be removed, as the histone mutants Htz1(H44C), Htz1(T49C), and Hta1(G38C) exhibited virtually no crosslinking. In fact, endogenous cysteines that are solvent exposed and in proximity to an interaction interface could potentially be exploited as cysteine probes for VivosX.

## Can the naturally occurring histone H3 C110 be used for VivosX?

A cysteine at position 110 (C110) of histone H3, which is conserved in most eukaryotes (but not in yeast), is within disulfide crosslinking distance with the same cysteine on the opposite H3 within nucleosomes (*Camerini-Otero and Felsenfeld, 1977*). However, to what extent the two nucleosomal

H3 proteins crosslink with each other via the C110-C110′ disulfide linkage remains unclear. Earlier studies on H3 C110 crosslinking encountered three technical challenges that are relevant to this study (*Garrard et al., 1977*). The first technical hurdle was the potential artifact of thiol-disulfide interchange after cell lysis (*Garrard et al., 1977*). This problem was overcome for VivosX by rapidly fixing the yeast cells with TCA before mechanical disruption and protein extraction in the presence of excess NEM, which effectively blocks non-specific disulfide crosslinking upon cell lysis (*Figure 1—figure supplement 3A–B*). Similarly, human proteins were directly extracted into NEM and the TCA fixation step was not necessary as the TUNES-NEM extraction buffer directly lysed the human cells. The second challenge was the preservation of the disulfide crosslinks formed inside cells during the protein extraction process (*Garrard et al., 1977*). For H2A and H2A.Z, no decrease in L1-L1′ adducts was observed as a function of time after lysis, suggesting that the L1-L1′ disulfide crosslinks are relatively stable under our experimental conditions (*Figure 1—figure supplement 3C–D*). The final challenge for the H3 studies was that the biological role of C110-C110′ crosslinking is unknown, thereby precluding the use of genetics to verify the H3 crosslinking status. We exploited the H2A.Z turnover pathway to perturb the L1-L1′ interaction of H2A.Z using mutants that both decrease and increase H2A.Z at promoters to validate the VivosX approach. Interestingly, in the VivosX analysis of human H2A.Z where efficient H2A.Z L1-L1′ crosslinking was observed, no 4-DPS-dependent C110-C110′ H3 crosslinking adduct was detected in the non-reducing anti-H3 immunoblot (*Figure 6—figure supplement 1A*). This is consistent with a previous conclusion that the C110 of histone H3 is inaccessible to thiol-specific reagents and requires destabilization of the nucleosome structure to make C110 solvent accessible (*Johnson et al., 1987*).

## VivosX can detect both modified and unmodified ZZ nucleosomes in human cells

The Cα-Cα′ distance for the L1-L1′ interface at H43 of human H2A.Z is 8.2 Å, which is 2.6 Å greater than that of the average disulfide bond (*Schmidt et al., 2006*; *Suto et al., 2000*). The ability of H2A.Z(H43C) to efficiently form intra-nucleosomal crosslinks suggests that the L1-L1′ interface of ZZ nucleosomes in human cells is not only solvent accessible, but also dynamic as in the case of the yeast ZZ nucleosomes. Unlike yeast, however, a subpopulation of H2A.Z in mammalian cells is mono-ubiquitylated, and these modified H2A.Z molecules are associated with facultative heterochromatin (*Sarcinella et al., 2007*). The observation that monoubiquitylated H2A.Z(H43C) readily forms disulfide crosslinks further supports that the accessibility and dynamics of the H2A.Z L1-L1′ interface is independent of transcription.

Our VivosX data suggest that up to two ubiquitin moieties can be simultaneously attached to one ZZ nucleosome. We deduce that the two ubiquitin moieties are separately attached to the opposite faces of the ZZ nucleosome given that reduction of the disulfide bond gave rise to mono- but not di-ubiquitylated H2A.Z. This observation is in concordance with a previous result that showed only one of the three lysines in the 120-**KK**GQQ**K**-125 motif at the C-terminus of H2A.Z was exclusively ubiquitylated (*Sarcinella et al., 2007*). Whether ZZ nucleosomes with one or two ubiquitin moieties represent biologically distinct chromatin states is unknown. In addition, cells with a similar ectopic *H2AFZ* vector (with an identical promoter but without the V5 tag and cysteine mutation) expressed ~5 times more protein than endogenous H2A.Z (*Figure 6—figure supplement 2*). The higher level of H2A.Z could cause misincorporation and thus the observed unmodified and ubiquitylated H2A.Z levels do not necessarily reflect the *in vivo* distribution of H2A.Z in the euchromatic and heterochromatic regions.

## Applications of VivosX beyond histone dynamics

The application of VivosX is not only limited to probing intra-nucleosomal histone-histone interactions, but also other protein-protein interactions where structural information is available. For example, VivosX could be used to elucidate the combinatorial interactions of basic helix-loop-helix leucine zipper (bHLH-LZ) family transcription factors. Using crystallographic data available for the bHLH-LZ heterodimers, like Myc-Max and Mad-Max, cysteine probes can be placed near the dimerization interface (*Nair and Burley, 2003*). As pairing of different bHLH-LZ dimers and subsequent binding to DNA sites are known to elicit specific cellular responses (*Diolaiti et al., 2015*), the relative

abundance of the disulfide adducts between the different bHLH-LZ partners could be used as a metric of pairing status in cells and analyzed by VivosX in a spatiotemporal manner.

The observation that ZZ nucleosomes can be monoubiquitylated asymmetrically on one H2A.Z or symmetrically on both raises the possibility that VivosX can be used to determine the stoichiometry of post-translational modifications (especially ubiquitin or SUMO) on other non-histone dimers or oligomers. For example, the DNA replication processivity clamp, proliferating cell nuclear antigen (PCNA), which functions as a trimer, can be conjugated to either ubiquitin or SUMO on the same lysine (*Choe and Moldovan, 2017*). Ubiquitylation promotes translesion synthesis bypass, whereas SUMOylation suppresses unwanted homologous recombination events (*Choe and Moldovan, 2017*). Whether these modifications are present on the same PCNA trimer *in vivo* is a question that can be addressed by VivosX. But in this case, two cysteine probes will be required per PCNA subunit since opposite sides of the molecule contribute to each interaction interface, and a DNA fragmentation step will be necessary to liberate the topologically linked PCNA after crosslinking (*Krishna et al., 1994*).

Although VivosX is not proven to detect protein-protein interactions in the cytosol, others have shown that cysteine probes on a redox sensitive green fluorescent protein can be induced to form intra-molecular disulfide bonds by 4-DPS treatment under experimental conditions similar to ours (*Hu et al., 2008*; *Østergaard et al., 2004*). Therefore, VivosX should theoretically be applicable to detecting cytosolic protein-protein interactions.

In summary, this work demonstrates that VivosX can reliably measure H2A and H2A.Z occupancy in yeast on the basis of disulfide crosslinking of cysteine probes located at the nucleosomal L1-L1' interface. When applied to human cells, VivosX revealed the complex nucleosomal configurations of H2A.Z that would be difficult to detect using traditional methods such as ChIP. Overall, VivosX is a simple but powerful strategy to capture site-specific, protein-protein interactions in cells that is applicable from yeast to humans.

## Materials and methods

### Yeast strains

The genotypes of the yeast strains used are listed in *Supplementary file 1*. The yeast strains used for yeast H2A.Z VivosX were generated by recombining the *HTZ1(Cys)2xFLAG-URA3* gene fragments at the original *HTZ1* locus of the *htz1Δ::kanMX* (1703) strain (*GE Dharmacon*). The *HTZ1(Cys)−2xFLAG-URA3* fragments, which contain flanking sequences at the 5' and 3' untranslated sequences of *HTZ1*, were polymerase chain reaction (PCR) amplicons that used *HTZ1(Cys)−2xFLAG-URA3* containing plasmids generated by site-directed mutagenesis of the *URA3 CEN ARS HTZ1-2xFLAG* plasmid (pEL353) as templates (*Wu et al., 2005*). The integrity of the resulting strains (yEL242-248) was verified by colony PCR and DNA sequencing of the amplified fragments using *Genewiz*.

The strains for H2A VivosX experiments were derived from the parental strain FY406, which lacks both copies of the endogenous genes for H2A and H2B and is complemented by a *URA3 CEN ARS HTA1-HTB1* plasmid (pSAB6) (*Hirschhorn et al., 1995*). The FY406 cells were then transformed with a *HIS3 CEN ARS 2xV5-HTA1-HTB1* plasmid (pEL305) or by its mutant variants containing the various cysteine substitutions in L1 of *HTA1*. Transformants were selected on the complete supplement mixture (CSM) medium lacking both histidine and uracil and then seeded onto medium without histidine but with uracil and 5-FOA to kill cells carrying the pSAB6 plasmid. The resulting yeast strains (yEL284, yEL285, yEL286, yEL489, yEL490, yEL491, and yEL492) carry the *HIS3 CEN ARS $^{V5}$HTA1(Cys)-HTB1* plasmids as the sole H2A source (Table 1—source data 1). The $^{V5}$*HTA1(N39C) HTZ1(T46C)$^{FL}$* haploid, yEL349, was derived from yEL286, which contains the $^{V5}$*HTA1(N39C)* allele. The *HTZ1(T46C)$^{FL}$-URA3* fragment was used to replace the endogenous *HTZ1* locus by homologous recombination.

The *swc5* mutant strains used in *Figure 4* were generated by transforming a *swc5Δ HTZ1(T46C)$^{FL}$* strain (yEL400) with *CEN ARS URA3* vectors bearing different *SWC5* mutants [pEL468: pRS416-swc5(P233*) and pEL479: pRS416-swc5(LDW→AAA)]. To construct yEL400, the *HTZ1(T46C)$^{FL}$-URA3* fragment was substituted for the endogenous *HTZ1* locus of the *swc5Δ* strain (3371, *GE Dharmacon*) generating yEL399. The *URA3* marker was then removed by homologous recombination with a 72

bp PCR amplicon linking the 2xFLAG tag region to the endogenous 3' untranslated region of *HTZ1*. The loss of the *URA3* marker was selected on medium containing 5-FOA. The strains used to conditionally deplete TBP and Swc5 (yEL402 and 403, respectively) and the untagged control (yEL401) were constructed by introducing the *HTZ1(T46C)^{FL}-URA3* cassette into the parental strains *TBP-FRB* (yEL098), *SWC5-FRB* (yEL054), and the untagged yEL044 strains, which contain the *tor1-1, fpr1Δ, RPL13A-2xFKBP12* alleles required for Anchor-away (*Haruki et al., 2008*; *Tramantano et al., 2016*).

## Plasmids

Descriptions of the plasmids and lentiviral vectors used in this study are listed in *Supplementary file 2*. The *CEN ARS URA3 HTZ1(Cys)^{FL}* plasmids, pEL427-432, were generated by site-directed mutagenesis using pEL353 (*CEN ARS URA3 HTZ1^{FL}*) as template (*Wu et al., 2005*). To add two V5 tags to the 5' end of the *HTA1* gene in the *CEN ARS HIS3 HTA1-HTB1* plasmid (JH55) (*Hirschhorn et al., 1995*), the plasmid was cut at two sites with *XbaI*, releasing a 225 bp fragment between −172 bp and +53 bp of *HTA1*. A synthetic gene fragment of the same region but with a 2xV5 tag immediately after the *HTA1* start codon was subcloned into the JH55 backbone via the *XbaI* sites to make pEL305. The *CEN ARS HIS3 ^{V5}HTA1(N39C)-HTB1* plasmid (pEL440), was generated by site-directed mutagenesis of pEL305 template. For the other *^{V5}HTA1(Cys)-HTB1* plasmids, pEL305 was linearized with BstAPI and MfeI, and synthetic double-stranded DNA fragments (*GenScript*) containing individual cysteine substitution at L1 with at least 60 bp of overlapping regions on both sides were recombined using the Gibson assembly kit (*New England Biolabs*), creating pEL558-pEL561. The integrity of all DNA constructs was verified by DNA sequencing.

## VivosX assay for yeast

The VivosX assay for yeast was developed based on a procedure described in *Dardalhon et al., 2012*. Yeast cells bearing the cysteine substituted L1 mutants (or the wild-type control) were cultured in 5 mL of CSM media (without cysteine, *Sunrise Science Products*) in 50-mL conical tubes at 30°C to logarithmic phase [Optical density, (OD)$_{600}$ of 0.5] before addition of 180 μM 4-DPS (*Sigma Aldrich*, Cat # 143057, 180 mM stock concentration in DMSO). Note that culturing cells in the Yeast Extract Peptone Dextrose (YEPD) media is not recommended for VivosX, as crosslinking efficiency is lower than when CSM media was used (data not shown), possibly due to neutralization of 4-DPS by free thiols in the YEPD media. The cultures were incubated at 30 °C for 20 min and then quenched with 20% TCA. After centrifugation at 2095 xg for 5 min in a swinging bucket rotor (SX4750, *Beckman coulter*), the pelleted cells were washed once with 1 mL 20% TCA and transferred to 1.7 mL screw cap tubes. After pelleting the cells again and removing the supernatant, homogenization was performed in the presence of 400 μL 20% TCA and ~450 μL of zirconia beads (0.7 mm diameter, *BioSpec*, 11079107zx) using a FastPrep-24 instrument (*MP Biomedicals*) at power level '6' for two times 20 sec with a 1 min incubation on ice in-between. After transferring 200 μL of the lysate to a new microcentrifuge tube, the precipitated total proteins along with cell debris were pelleted by centrifugation at 20,400 xg for 15 min at 4 °C. The pellets were then washed with ice-cold acetone, minced with pipet tips, and dispersed as much as possible. The insoluble materials were pelleted again by centrifugation at 20,400 xg for 15 min at 4 °C before re-suspending in the TUNES buffer (100 mM Tris pH 7.2, 6 M Urea, 10 mM EDTA, 1% SDS, 0.4 M NaCl, 10% glycerol). Where indicated, 50 mM of NEM was added. Extraction was performed at 30°C for 60 min with constant mixing in a vortexer (*TOMY*, MT-400) followed by centrifugation at 20,400 xg for 10 min. For non-reducing SDS-PAGE, 1 part of 1% bromophenol blue was added to 24 parts of the cleared extracts. For reducing SDS-PAGE, 25 parts of the bromophenol blue/extract mixtures were mixed with 1 part of β-ME and heated at 55 °C for 5 min. Electrophoresis was performed at 150 V for 1 hr 25 min in the Tris-Glycine-SDS running buffer (25 mM Tris, 192 mM glycine, 0.1% w/v SDS). The Prestained Precision Plus Protein Standards (*BioRad*, Cat. 1610375) were used in all gels.

The electrophoresed proteins were transferred to methanol-treated PVDF membranes (*BioRad*, Cat # 162–0174) using the XCell II blotting system at a constant voltage of 25 V for 2 hr in the Transfer buffer (12.4 mM Tris, 96 mM glycine, 0.1% w/v SDS, 20% v/v methanol). The membranes were blocked with 2% of the ECL Prime Blocking agent (*GE Healthcare*) and probed with the indicated antibodies. Anti-FLAG (*Sigma Aldrich*, Cat. F3165) was used at a dilution of 1:2000, anti-H2A (*Active Motif*, Cat. 39235) at 1:2000, anti-H3 (gift of Carl Wu) at 1:2000, anti-V5 (*Thermo Fisher*, Cat. MA5-

15253) at 1:2000, and anti-human H2A.Z (Active Motif, Cat. 39943) at 1:2000. Normalization of loading of gels was performed empirically. This was achieved by an initial immunoblotting analysis that uses equal volume (25 μL) of samples followed by a second analysis that uses adjusted volumes (10–40 μL) based on the signals [i.e. anti-FLAG for Htz1(Cys)$^{FL}$ and anti-H2A for $^{V5}$Hta1(Cys)] of the β-ME-treated samples of the first immunoblot.

Preparation of the cultures of the Anchor-away strains was modified for VivosX as follows. *TBP-FRB* (yEL402), *SWC5-FRB* (yEL403) and untagged control (yEL401) strains were cultured in 50 mL CSM medium to an $OD_{600}$ of 0.5 before the addition of 1 ug/mL rapamycin (RAP, 1 mg/mL stock concentration in DMSO) or equal volume of DMSO. After incubation at 30°C for 1 hr, 5 mL cultures were transferred into a 50-mL falcon tube and incubated with either 180 μM 4-DPS or equivalent volume of DMSO for 20 min. Protein extraction and immunoblotting were performed as described above. In the experiment where azide was used to block cellular activities, 0.1% of sodium azide was added (*Bermejo et al., 2007*) in conjunction with 4-DPS and incubated for 20 min before TCA fixation.

## Disulfide crosslinking of native nucleosomes

Native nucleosomes were prepared from cells harvested from 400 mL logarithmic growing ($OD_{600}$ = 0.5) cultures of yEL314, yEL356 and yEL349 (see Table 1—source data 1) in CSM medium. The cells were harvested by centrifugation and washed with 1x phosphate buffered saline (PBS) before they were spheroplasted with lyticase (L2524-50KU, *Sigma-Aldrich*) as described in (*Tramantano et al., 2016*). Spheroplasts were lysed in a Dounce homogenizer using 350 μL of extraction buffer [50 mM HEPES (pH 7.5), 80 mM NaCl, 0.25% Triton X-100, 0.17 mg/mL PMSF, 0.33 mg/mL benzamidine hydrochloride, 13.7 μg/mL pepstatin A, 0.284 μg/mL leupeptin, 2 μg/mL chymostatin]. After centrifugation at 9100 x g for 10 min at 4°C, the crude chromatin, which is in the pellet fraction, was resuspended using 150 μL of the extraction buffer containing 1 mM $CaCl_2$. Nucleosomes were liberated from the insoluble chromatin by digestion with MNase (12 units, *Worthington*) at 37 °C for 5 min. The MNase reactions were quenched by 10 mM EGTA. The soluble fraction containing the nucleosomes was cleared by passing through a 0.22 μm PVDF membrane (*Millipore*, Cat UFC40GV0S). Fifty microliters of nucleosomes was incubated with or without 180 μM 4-DPS at 30°C for 20 min. The histones were then precipitated with 20% TCA and pelleted by centrifugation at 20,400 x g for 5 min. The histone proteins were re-solubilized using the NEM-containing TUNES buffer and analyzed by SDS-PAGE and immunoblotting as describe above.

## Lentiviral vectors

The lentiviral vectors used in the expression of human *H2AFZ* and its mutant derivatives were constructed using the pLT-iGSP vector system (*Leung and Brugge, 2012*). Synthetic DNA fragments containing the coding sequences of human *H2AFZ*, *H2AFZ(H43C)* [CAT→TGT], *H2AFZ(KK→RR)* [AAGAAA→AGGAGA at amino acid sequence 121 and 122] or *H2AFZ(H43C, KK→RR)* (*GenScript* or *Integrated DNA Technologies*) with Kozak sequence, gccaccAUG, and an in-frame C-terminal 2xV5 tag were inserted into pLT-iGSP via the *XbaI* and *BamHI* sites. The *H2AFZ$^{V5}$* construct and its mutant variants were placed under the control of the tetracycline-response element (TRE) and fused 3' to an IRES (internal ribosomal entry site)-mediated bicistronic green fluorescent protein (GFP) reporter (*Leung and Brugge, 2012*). A constitutive expression cassette of the puromycin N-acetyltransferase gene in the pLT-iGSP was utilized for selection of stably-transduced cell clones (*Leung and Brugge, 2012*).

## Human cell culture and virus production

MCF10A (*ATCC*, CRL-10317) cells were cultured in Dulbecco's Modified Eagle Medium/Nutrient Mixture F-12 (DMEM/F-12) (*GE Healthcare*) supplemented with 5% horse serum (*Thermo Fisher*), 20 ng/ml epidermal growth factor (EGF) (*PeproTech*), 0.5 μg/ml hydrocortisone, 0.1 μg/ml cholera toxin, 10 μg/ml insulin (*Sigma Aldrich*), 50 U/ml penicillin, and 50 U/ml streptomycin (*Thermo Fisher*). HEK293T (*ATCC*, CRL-3216) cells were cultured in Dulbecco's High Glucose Modified Eagles Medium supplemented with 10% fetal bovine serum (*Thermo Fisher*), 15 mM HEPES (*Sigma Aldrich*), 50 U/ml penicillin, and 50 U/ml streptomycin. All cells were cultured at 37°C and 5% $CO_2$.

For virus production, HEK293T cells were co-transfected with each lentiviral vector together with the packaging vectors, psPAX2 and pMD2.G (*Addgene*) using the TurboFect transfection reagent (*Thermo Fisher*). Viruses were collected on day 3 and 4 post-transfection and filtered through a 0.45 μm membrane. The stable cell lines used in *Figure 6* were generated by infecting MCF10A cells that carry the reverse tetracycline transactivator (MCF10A/pBABE-hygro-rtTA) (*Leung and Brugge, 2012*) with the lentiviruses described above and selected with 2 μg/ml puromycin (*Thermo Fisher*). The parental MCF10A cells were authenticated using the *ATCC* Human STR Profiling Cell Authentication Service.

## VivosX assay for human cells

Approximately, 100,000 cells were seeded on each well of a 6-well tissue culture plate. One day after seeding, the cells were treated with 1 μg/ml doxycycline for 48 hr to induce the expression of the *H2AFZ*$^{V5}$ gene and its mutant variants. Induced cells were treated with 180 μM of 4-DPS or vehicle control DMSO for 20 min at 37°C and cell lysates were collected using the TUNES buffer with 50 mM NEM and 1X Halt Protease and Phosphatase Inhibitor Cocktail (*Thermo Fisher*). To fragment the genomic DNA before gel loading, the lysates were sonicated for a total 5 min (30 cycles of 10 s ON and 10 s OFF) in an ice-chilled water bath of a 5.5-inch inverted cup horn probe with 60% power using a 700 watts processor (Q700, *Qsonica*). One part of 1% bromophenol blue was added to 24 parts of the sonicated lysate with or without β-ME (1 part) before SDS-PAGE and immunoblotting analysis, which were performed as described for the yeast VivosX experiments.

# Acknowledgements

This work was inspired by insightful discussions with Rolf Sternglanz, Caryn Outten, and Steven Glynn. We thank Nancy Hollingsworth and Erwin London for critical comments of the manuscript, members of the Luk laboratory for helpful suggestions, Fred Winston for sharing the *HTA1-HTB1* plasmid (JH55) and the yeast strain (FY406) and Carl Wu for sharing the anti-H3 antibody. This work is supported by research grants from the National Institutes of Health (RO1 GM104111 to EL and RO1 CA200652 to CTL).

# Additional information

## Funding

| Funder | Grant reference number | Author |
|---|---|---|
| National Institute of General Medical Sciences | RO1 GM104111 | Ed Luk |
| National Cancer Institute | RO1 CA200652 | Cheuk T Leung |

The funders had no role in study design, data collection and interpretation, or the decision to submit the work for publication.

## Author contributions

Chitra Mohan, Ed Luk, Conceptualization, Resources, Data curation, Formal analysis, Supervision, Funding acquisition, Validation, Investigation, Visualization, Methodology, Writing—original draft, Project administration, Writing—review and editing; Lisa M Kim, Cheuk T Leung, Conceptualization, Data curation, Formal analysis, Supervision, Funding acquisition, Validation, Investigation, Methodology, Project administration, Writing—review and editing; Nicole Hollar, Tailai Li, Data curation, Formal analysis, Validation, Investigation, Methodology, Writing—review and editing; Eric Paulissen, Data curation, Formal analysis, Investigation, Methodology, Writing—review and editing

## Author ORCIDs

Ed Luk http://orcid.org/0000-0002-6619-2258

## Decision letter and Author response

Decision letter https://doi.org/10.7554/eLife.36654.023

Author response https://doi.org/10.7554/eLife.36654.024

---

## Additional files

### Supplementary files

• Supplementary file 1. Table of yeast strains used in this study.
DOI: https://doi.org/10.7554/eLife.36654.019

• Supplementary file 2. Table of plasmids and lentiviral vectors used in this study
DOI: https://doi.org/10.7554/eLife.36654.020

• Transparent reporting form
DOI: https://doi.org/10.7554/eLife.36654.021

### Data availability

Source data files of western blotting quantification are included as Figure supplements.

---

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
