## [Decision Letter]

Thank you for submitting your article "Using VivosX, a disulfide crosslinking approach, to capture intra-nucleosomal interactions in yeast and human cells" for consideration by *eLife*. Your article has been reviewed by three peer reviewers, including Geeta J Narlikar as the Reviewing Editor and Reviewer #1, and the evaluation has been overseen by Kevin Struhl as the Senior Editor. The following individual involved in review of your submission has agreed to reveal their identity: Tim Formosa (Reviewer #2).

The reviewers have discussed the reviews with one another and the Reviewing Editor has drafted this decision to help you prepare a revised submission.

Summary:

In this "Tools and Resources" manuscript the authors describe a novel disulphide cross-linking approach that they term VivosX to assess in vivo interactions between canonical and variant H2A histones. The approach relies on strategically placed cysteines that are expected to form disulphide bonds under oxidizing conditions. The authors provide extensive validation of the method and clear description of its use. While they note that the results obtained must be considered to be qualitative assessments, they demonstrate that both global decreases and increases in H2A.Z-containing nucleosomes that result from physiological perturbations were detectable using their VivosX technique. For example, in previous work the Luk group showed that PIC formation is responsible for evicting H2A.Z-containing nucleosomes, and here they find that PIC preferentially removes homo-ZZ over hetero-ZA nucleosomes. They also show that a ubiquitin modification at the C-terminus can be detected for H2A.Z nucleosomes. Overall the VivosX approach appears to be a useful approach to rapidly screen for effects of mutants and environmental conditions on the composition of nucleosomes in terms of H2A and H2A.Z. A strength of the approach is its relative ease of use because it relies on directly probing histone proteins that are TCA precipitated from whole cell extracts.

*eLife*'s policy on this category of papers states that new methods "will be assessed in terms of their potential to facilitate experiments that address problems that to date have been very challenging or even intractable." On this point the authors succeed, as the difficulty of measuring changes in H2A.Z-containing nucleosomes has posed an impediment to understanding the roles of this histone variant. The category description also states that "Tools and Resources articles do not have to report major new biological insights or mechanisms, but it must be clear that they will enable such advances to take place." It is less clear that, as written, this criterion is met; the authors don't discuss how this new method might be applied to address questions beyond those relating to the H2A/H2A.Z composition of nucleosomes. Below we suggest some ways in which the authors can discuss the broader uses of the VivosX approach.

1) With the numerous hetero and homo-dimeric complexes in the cell (where crystal structures are known), the authors' work makes it a realistic goal to monitor the relative proportions of these pairings by engineering S-S bonds (for example, Myc-Max and Mad-Max and there are many others). The authors should discuss such possibilities.

2) The authors can use their ubiquitylated H2A.Z results to discuss how VivosX can assess whether non-histone protein dimers have symmetric or asymmetric post-translational modifications.

3) The authors can discuss if it's possible to estimate the amount of free or unincorporated H2A.Z in cells, based on the mono Z bands in the gel. The authors mention several reasons why cross-linking might not occur. However recent work from the Richmond lab (https://doi.org/10.1016/j.jmb.2017.10.029) suggests that the corresponding S-S L1 loop cross-link in H2A is quite efficient within a nucleosome *in vitro*. The authors could discuss how variations in cross-linking efficiency may be reflective of nucleosomal vs. non-nucleosomal H2A.Z (bound to SWR1 or chaperones) or perhaps H2A.Z in hexasomes. The same could hold for H2A.

Overall rewriting with a broader view would make VivosX more appropriate as a general vs. nucleosome-specific tool.

There are additional issues that the authors should address as described below.

Essential revisions:

1) The data in Figure 5C is the key new finding being proposed using this method. While the data appears qualitatively consistent with the authors' conclusion, quantification of the relative amounts of H2A.Z-H2A.Z vs. H2A.Z-H2A cross-links normalized to mono Z is essential. In addition, error bars are needed.

2) The authors should explain why the residue corresponding to R48 in yeast Htz1 was used for the human studies instead of the T46 residue used for the yeast method?

3) How was normalization of loading for gels performed; a constant fraction of the sample normalized by cell number or some other parameter?

[Editors' note: further revisions were requested prior to acceptance, as described below.]

Thank you for resubmitting your work entitled "VivosX, a disulfide crosslinking approach to capture site-specific, protein-protein interactions in yeast and human cells" for further consideration at *eLife*. Your revised article has been favorably evaluated by Kevin Struhl (Senior Editor), a Reviewing Editor, and three reviewers.

The manuscript has been much improved by rewriting and expanding. The reviewers felt that some additional minor reorganization would help better convey the significance of this work as a general methodology.

1) In the Discussion section, it isn't until the very end that the general applications are brought up. The following rearrangements would help: Move the first to fourth paragraphs of subsection “Applications of VivosX beyond histone dynamics” to the beginning of the Introduction. Then the proof of principle of H2A.Z can be introduced as an example after that. The final paragraph of the Discussion section would still be a nice recap in its position as last paragraph.

2) In the Introduction, when first mentioning VivosX, it would be helpful to say something like "Here we describe/introduce a methodology we call VivosX" to make it clear that it is new and the point of what is to follow.

3) The last paragraph of the Introduction needs to be reworded. It is presented as a continuum with the intro material. The conclusions are given before the system is really set up, so it feels like it gets ahead of itself. The first few sentences are all about H2A.Z, so that seems to be the main point of the paper (as was the previous version). Instead, it would be better to begin with the idea of VivosX as a general strategy and presenting H2A.Z as a test case that works. The statements about the generality of VivosX come too late in the paragraph and it does not flow well. Flipping the paragraph around and starting with VivosX will help. Also, it would be useful to make a distinction between background information and what is being presented from this novel work. Therefore, in the Introduction, where the statement is made "Single cysteine substitutions were systematically introduced", it would be helpful to prefix something like "In this study" or "Here we" to let the reader know you are now describing what the paper is about.

4) The authors discuss the relevance of finding ubiquitylated H2A.Z in human cells well in later parts of the manuscript, but as this result is mentioned in the Abstract and Introduction, some context for why this outcome is important should be provided earlier.

5). The effects in Figure 5C are small, though within error. The authors should state this explicitly in the text (subsection “ZZ nucleosomes are preferentially disassembled by the PIC”) by saying something like: "While the increase in the ZxZ adduct and the corresponding decrease in monomeric H2A.Z was small, the effects were reproducible within error".

---

## [Author Response]

Summary:

*In this "Tools and Resources" manuscript the authors describe a novel disulphide cross-linking approach that they term VivosX to assess* in vivo *interactions between canonical and variant H2A histones. The approach relies on strategically placed cysteines that are expected to form disulphide bonds under oxidizing conditions. The authors provide extensive validation of the method and clear description of its use. While they note that the results obtained must be considered to be qualitative assessments, they demonstrate that both global decreases and increases in H2A.Z-containing nucleosomes that result from physiological perturbations were detectable using their VivosX technique. For example, in previous work the Luk group showed that PIC formation is responsible for evicting H2A.Z-containing nucleosomes, and here they find that PIC preferentially removes homo-ZZ over hetero-ZA nucleosomes. They also show that a ubiquitin modification at the C-terminus can be detected for H2A.Z nucleosomes. Overall the VivosX approach appears to be a useful approach to rapidly screen for effects of mutants and environmental conditions on the composition of nucleosomes in terms of H2A and H2AZ. A strength of the approach is its relative ease of use because it relies on directly probing histone proteins that are TCA precipitated from whole cell extracts.*

The reviewers’ comments make us realize that our comment about VivosX being qualitative is inaccurate, as this approach was used to make quantitative assessments, such as those found in Figures 5A and 5C. Therefore, the sentence ‘the VivosX readout is qualitative’ is removed.

eLife's policy on this category of papers states that new methods "will be assessed in terms of their potential to facilitate experiments that address problems that to date have been very challenging or even intractable." On this point the authors succeed, as the difficulty of measuring changes in H2A.Z-containing nucleosomes has posed an impediment to understanding the roles of this histone variant.

Thanks!

The category description also states that "Tools and Resources articles do not have to report major new biological insights or mechanisms, but it must be clear that they will enable such advances to take place." It is less clear that, as written, this criterion is met; the authors don't discuss how this new method might be applied to address questions beyond those relating to the H2A/H2A.Z composition of nucleosomes. Below we suggest some ways in which the authors can discuss the broader uses of the VivosX approach.

To emphasize the general application of VivosX, we have revised the manuscript as follows:

First, the title has been changed to “VivosX, a disulfide crosslinking approach to capture site-specific, protein-protein interactions in yeast and human cells”. Second, the Introduction was revised to explicitly state that VivosX is a general strategy for probing site-specific, protein-protein interactions and that the histone study is a proof of concept for the new approach. Third, the application of VivosX to study questions beyond histone dynamics is discussed (Subsection “Applications of VivosX beyond histone dynamics”).

1) With the numerous hetero and homo-dimeric complexes in the cell (where crystal structures are known), the authors' work makes it a realistic goal to monitor the relative proportions of these pairings by engineering S-S bonds (for example, Myc-Max and Mad-Max and there are many others). The authors should discuss such possibilities.

As mentioned above, potential applications of VivosX for probing other protein–protein interactions are discussed. The Myc-Max/Mad-Max example is highly

relevant and we are grateful to the reviewer for suggesting it. (subsection “Applications of VivosX beyond histone dynamics”).

2) The authors can use their ubiquitylated H2A.Z results to discuss how VivosX can assess whether non-histone protein dimers have symmetric or asymmetric post-translational modifications.

We now discuss how VivosX can be used to determine the stoichiometry of posttranslational modifications on dimers and oligomers. (subsection “Applications of VivosX beyond histone dynamics”).

*3) The authors can discuss if it's possible to estimate the amount of free or unincorporated H2A.Z in cells, based on the mono Z bands in the gel. The authors mention several reasons why cross-linking might not occur. However recent work from the Richmond lab (https://doi.org/10.1016/j.jmb.2017.10.029) suggests that the corresponding S-S L1 loop cross-link in H2A is quite efficient within a nucleosome* in vitro*. The authors could discuss how variations in cross-linking efficiency may be reflective of nucleosomal vs. non-nucleosomal H2A.Z (bound to SWR1 or chaperones) or perhaps H2A.Z in hexasomes. The same could hold for H2A.*

We have estimated the amount of unpaired A-B and Z-B dimers to be 39% and 26%, respectively. We now discuss the source of these unpaired histones in a new paragraph starting at Line 502. The work from the Richmond lab regarding the crosslinking efficiency of H2A L1-L1’ is cited therein.

Overall rewriting with a broader view would make VivosX more appropriate as a general vs. nucleosome-specific tool.

The revised text now emphasizes that VivosX is a general tool for probing protein–protein interactions.

There are additional issues that the authors should address as described below.Essential revisions:1) The data in Figure 5C is the key new finding being proposed using this method. While the data appears qualitatively consistent with the authors' conclusion, quantification of the relative amounts of H2A.Z-H2A.Z vs. H2A.Z-H2A cross-links normalized to mono Z is essential. In addition, error bars are needed.

Quantification of the averages for three biological replicates of Figure 5B (previously 5C, as a model figure has been moved to Figure 1 figure supplement 1) with error bars is now included as a new Figure 5C. However, we normalized the AxZ and ZxZ data to total H2A.Z instead of monomer H2A.Z (mono Z) because mono Z level is reciprocal to AxZ and ZxZ levels. This is problematic for statistical analysis as the denominator is not an independent variable of the numerator.

2) The authors should explain why the residue corresponding to R48 in yeast Htz1 was used for the human studies instead of the T46 residue used for the yeast method?

H2A.Z(H43C), which corresponds to the yeast Htz1(R48C) was used in the human studies because the crosslinking efficiency of Htz1(R48C) was slightly better than Htz1(T46C) in our preliminary experiments, although the crosslinking enhancement was later found to be insignificant. This explanation can be found in subsection “ZZ nucleosomes are preferentially disassembled by the PIC”.

3) How was normalization of loading for gels performed; a constant fraction of the sample normalized by cell number or some other parameter?

Normalization of loading of gels was performed empirically. This was achieved by an initial immunoblotting analysis that uses equal volume (25 μL) of samples followed by a second analysis that uses adjusted volumes (10-40μL) based on the histone signals of the bME-treated samples of the first immunoblot. (Subsection “VivosX assay for yeast”).

[Editors' note: further revisions were requested prior to acceptance, as described below.][…] 1) In the Discussion section, it isn't until the very end that the general applications are brought up. The following rearrangements would help: Move the first to fourth paragraphs of subsection “Applications of VivosX beyond histone dynamics” to the beginning of the Introduction. Then the proof of principle of H2A.Z can be introduced as an example after that. The final paragraph of the Discussion section would still be a nice recap in its position as last paragraph.

We agree that the general applications for VivosX should be brought up earlier in the Introduction. However, we feel that bringing up specific examples, such as Myc and Max, in the Introduction would create false expectations that VivosX would be applied to these proteins in the current study. Instead of highlighting specific nuclear factors, we say in the first paragraph of the Introduction that VivosX can be used to study how transcription factors or signaling molecules dimerize in response to cellular cues. We then say that histone crosslinking was used to test the robustness of the assay. The application of using VivosX to determine the stoichiometry of post-translational modifications on individual subunits of an oligomer is brought up later in the introduction after stating that VivosX can be used to reveal the ubiquitylation status of H2A.Z nucleosomes in human cells.

2) In the Introduction, when first mentioning VivosX, it would be helpful to say something like "Here we describe/introduce a methodology we call VivosX" to make it clear that it is new and the point of what is to follow.

Agreed – this change has been made.

3) The last paragraph of the Introduction needs to be reworded. It is presented as a continuum with the intro material. The conclusions are given before the system is really set up, so it feels like it gets ahead of itself. The first few sentences are all about H2A.Z, so that seems to be the main point of the paper (as was the previous version). Instead, it would be better to begin with the idea of VivosX as a general strategy and presenting H2A.Z as a test case that works. The statements about the generality of VivosX come too late in the paragraph and it does not flow well. Flipping the paragraph around and starting with VivosX will help.

Agreed. We split the last paragraph of the original Introduction into two and reworded them to emphasize the idea that VivosX is a general methodology and that histone crosslinking is a test case.

Also, it would be useful to make a distinction between background information and what is being presented from this novel work. Therefore, in the Introduction, where the statement is made "Single cysteine substitutions were systematically introduced", it would be helpful to prefix something like "In this study" or "Here we" to let the reader know you are now describing what the paper is about.

Agreed – this change has been made.

4) The authors discuss the relevance of finding ubiquitylated H2A.Z in human cells well in later parts of the manuscript, but as this result is mentioned in the Abstract and Introduction, some context for why this outcome is important should be provided earlier.

The idea that H2A.Z-containing nucleosomes with and without ubiquitylation are associated with different regions of the genome is now brought up earlier in the Introduction.

5). The effects in Figure 5C are small, though within error. The authors should state this explicitly in the text (subsection “ZZ nucleosomes are preferentially disassembled by the PIC”) by saying something like: "While the increase in the ZxZ adduct and the corresponding decrease in monomeric H2A.Z was small, the effects were reproducible within error".

This section is reworded to explicitly state that the differences were small but reproducible, with non-overlapping standard deviations and p values of < 0.05 based on t-test. (Subsection “ZZ nucleosomes are preferentially disassembled by the PIC”).